# (Re)Emerging Arboviruses of Public Health Significance in the Brazilian Amazon

**DOI:** 10.3390/microorganisms13030650

**Published:** 2025-03-12

**Authors:** Kyndall C. Dye-Braumuller, Rebecca A. Prisco, Melissa S. Nolan

**Affiliations:** 1Department of Epidemiology and Biostatistics, Arnold School of Public Health, University of South Carolina, Columbia, SC 29208, USA; rprisco@email.sc.edu (R.A.P.); msnolan@mailbox.sc.edu (M.S.N.); 2Institute for Infectious Disease Translational Research, Arnold School of Public Health, University of South Carolina, Columbia, SC 29208, USA

**Keywords:** Brazil, arbovirus, emerging, re-emerging, arthropod vectors, ecology deforestation, urbanization

## Abstract

Brazil is one of the most important countries globally in regard to arboviral disease ecology and emergence or resurgence. Unfortunately, it has shouldered a majority of arboviral disease cases from Latin America and its rich flora, fauna (including arthropod vectors), and climate have contributed to the vast expansion of multiple arboviral diseases within its borders and those that have expanded geographically outside its borders. Anthropogenic landscape changes or human-mediated changes such as agriculture, deforestation, urbanization, etc. have all been at play within the country in various locations and can also be attributed to arboviral movement and resurgence. This review describes a brief history of landscape changes within the country and compiles all the known information on all arboviruses found within Brazil (endemic and imported) that are associated with human disease and mosquitoes including their original isolation, associated vertebrate animals, associated mosquitoes and other arthropods, and human disease symptomology presentations. This information is crucial as the Western Hemisphere is currently experiencing multiple arbovirus outbreaks, including one that originated in the Brazilian Amazon. Understanding which arboviruses are and have been circulating within the country will be pertinent as anthropogenic landscape changes are consistently being perpetrated throughout the country, and the occurrence of the next arbovirus epidemic will be a matter of when, not if.

## 1. Introduction

Arboviruses, viruses transmitted by arthropod vectors, such as mosquito-borne yellow fever (YFV) and dengue (DENV) viruses cause significant morbidity and mortality globally: the World Health Organization estimates over 400 million infections, 70,000 deaths, and that 3.9 billion people are at risk for both YFV and DENV disease annually [1,2]. Multiple arboviruses have caused global pandemics in recent years, including the Chikungunya (CHKV), Zika (ZIKV), and West Nile (WNV) viruses, for example [3,4]. This resurgence in geographical spread and disease incidence is considered one of the most significant public health threats in the Americas [4,5].

South America has a vast history of supporting vector-borne arboviral transmission, whether endemic or epidemic, vectored by multiple arthropods including mosquitoes and sand flies. This is due to multiple factors including the favorable climate (and subsequent climate change impacts on this), the availability of hosts and vectors, expanses in travel and commerce, and urban sprawl from human populations increasing in size and movement across and into new landscapes [3,6]. Brazil has shouldered a large proportion of the South American arbovirus transmission burden, including outbreaks of multiple imported and endemic arboviruses—many of which have spread throughout the continent and into North America, causing international outbreaks [7,8,9]—with its highly favorable climate and the Amazon rainforest supporting various cryptic and appropriate breeding habitats along with potential host animals; the large diversity of vector fauna (279 species of sand flies [10] and 516 species of mosquitoes) [11] reported in the country; high rates of anthropogenic change through urbanization, deforestation, pollution, etc.; and large, populous cities with trade ports contributing to human movement, trade, and the exposure of naïve individuals to new viruses [3,6,12].

### Brief History of Deforestation in the Brazilian Amazon

The deforestation of the Brazilian Amazon began in the late 1950s in favor of building transportation infrastructure; agricultural development for the beef, soy, and logging industries; and increasing human habitation (urbanization) [13,14,15]. Nine federal states cover the Brazilian Legal Amazon: Acre, Amapá, Amazonas, Maranhão, Mato Grosso, Pará (PA), Rondônia (RO), Roraima, and Tocantins. The resultant highways at the heart of this ‘arc of deforestation’ in the Brazilian Amazon include the Belém–Brasilia highway (BR-316, BR-308, and BR-010), integrating the western and northern states with the rest of the country (through Amapá, Pará, Maranhão, and Tocantins); the Cuiba–Porto Velho (BR-364) highway providing access to the southern region of the Amazon (through Acre, Amazonas, Rondônia, and Mato Grosso); the Transamazon (BR-230), running west to east through the Amazon (through Amazonas, Pará, and Tocantins); the BR-163, connecting the southernmost portion to the east (through Mato Grosso and Pará); and the BR-174, running from the northernmost state to the center of the Amazon (through Roraima and Amazonas) [14]. Recent deforestation efforts can be largely attributed to the cattle ranching industry more than transportation infrastructure [14,16]. Between 1996 and 2005, the average annual deforestation rate was 19,500 km^2^/year [17]. Despite declarations of reducing deforestation rates since the early 2000s, rates have slowed but have not reached the levels promised—largely due to political instability and economic recession [18,19]. In addition, the illegal appropriation of undesignated public lands by land grabbers has also contributed to occupation and subsequent smaller-scale deforestation, leading to increased clearing by these land grabbers [19]. By 2021, approximately 20% of the 4 million km^2^ originally forested portion of the Brazilian Legal Amazon had been cleared [19].

Deforestation and urbanization are two of the most important anthropogenic land-use changes known to impact infectious disease epidemiology [20,21]. The consequences of these include bringing human inhabitants into the deforested area (oftentimes bringing livestock), with resultant increased vector breeding habitats near novel vertebrate hosts, increased temperatures and fewer seasonal changes in a more urban environment, and the potential for socioeconomically weaker areas to emerge [20,21,22]. Increased evidence has emerged directly linking vectors and vector-borne disease to anthropogenic changes in the environment, specifically in Latin America: deforested and disturbed areas have led to the increased presence of mosquitoes and arbovirus outbreaks [23,24,25,26,27,28].

## 2. Arboviruses Associated with Mosquitoes and Humans in Brazil

Over 210 arboviruses have been isolated and reported in Brazil alone, and it is thought that 196 (93%) originated in Brazil, the majority of which were isolated specifically in the Brazilian Amazon [29,30]. Of these Brazilian arboviruses, 39 are associated (found through serological or molecular screening methods) with both humans and mosquitoes (Figure 1, Table 1). Association with humans and mosquitoes does not necessarily equate to transmission, viremia development, or disease development; however, the literature suggests that at least a majority of these 39 viruses are transmitted by the listed mosquitoes and can infect humans. Caution should also be taken when consuming information regarding serological test results from viruses as multiple viruses within the same family can cross-react with one another. Here, we summarize these medically relevant arboviruses’ ecologies grouped by viral families to enhance the scientific community’s knowledge of Brazilian mosquito-borne pathogens of (re)emergence potential, particularly including the lesser-known pathogens.

### 2.1. Togaviridae

The Togaviridae family is an important group of viruses globally. Of this group, nine in the genus *Alphavirus* have been isolated in the Brazilian Amazon and associated with mosquitoes and humans—many with either a global or multi-country geographical reach. All alphaviruses are single-stranded positive-sense RNA viruses associated with mosquitoes [31]. A general depiction of the *Togaviridae* transmission cycles and vertebrate animal associations is shown in Figure 2.

#### 2.1.1. Mayaro Virus (MAYV)

Mayaro virus (MAYV) was originally isolated in a febrile human man in Mayaro County, Trinidad in 1954 [32]. Transmission dynamics include a sylvatic cycle where the virus is circulated between non-human primates from primarily *Haemagogus* spp. mosquitoes, which primarily inhabit forested canopies [3,33,34]. Additional vertebrate hosts in this sylvatic environment include birds, rodents, sloths, and opossums; however, non-human primates are considered the most important vertebrate host [34]. Although urban outbreaks are not frequent, humans can become infected with MAYV; *Haemagogus* mosquitoes have been associated with human cases, specifically *Hg. janthinomys* [3,35], and *Aedes aegypti* mosquitoes are competent vectors in laboratory conditions [36]. Disease presents as an acute febrile illness with fever, arthralgia and arthritis, and a maculopapular rash [34]; less frequent additional symptoms include headache, myalgia, retro-orbital pain, vomiting, and diarrhea [37]. There is also evidence that recurring or persistent arthralgia can impact victims for months or years following initial infection [38,39,40].

In Brazil, major groups of mosquito species associated with MAYV isolations include *Haemagogus*, *Sabethes*, and *Culex* spp. (unknown species) [41]. The first documented outbreak of MAYV in Brazil was in Pará State, Amazonia: in Guamá, in 1955 [42]; multiple outbreaks have occurred in the Brazilian Amazon since, indicating consistent human exposure to MAYV [43,44,45]. A large arbovirus study from the 1950s to the 1960s in Pará State isolated MAYV from humans relatively quickly following this first outbreak and found evidence of antibodies to the virus in humans (60% of individuals tested) [41]. Mayaro virus outbreaks and evidence of infection through serological studies have also been documented in central and southern Brazil, notably in São Paulo, Mato Grosso, Tocantins, and Goiás States most recently, indicating viral spread outside of the Brazilian Amazon and the potential for epidemics in other parts of the country [46,47,48].

Recent circulation: Notably, an increase in the occurrence of MAYV cases has been seen in several Brazilian states with outbreaks, with even more cases potentially hidden by concurrent epidemics of a related virus, Chikungunya virus, circulating in the same vector species and ecological niche [46]. Multiple recent serological investigations have noted evidence of consistent MAYV exposure throughout the country [34].

#### 2.1.2. Chikungunya Virus (CHIKV)

Chikungunya virus (CHIKV) was first described in southern present-day Tanzania, having been the cause of several outbreaks of dengue-like disease; in 1953, the virus was successfully isolated from a febrile female patient [49]. In its native range in Africa, CHIKV has been detected in over 30 species of mosquitoes [50]. The enzootic cycle is circulated through arboreal *Aedes* spp. and non-human primates while the epidemic cycle is primarily driven by both *Ae. aegypti* and *Ae. albopictus* mosquitoes in urban locations [50]. In the Western Hemisphere, little is known regarding an enzootic transmission cycle. Arboreal mosquito species are potential sylvatic vectors, and multiple mosquito species in other genera have been found infected with CHIKV [51,52]; however, low seropositivity rates have been reported in South American non-human primates [50,53]. In contrast, both *Ae. aegypti* and *Ae. albopictus* are the primary urban transmission vectors of CHIKV in the Western Hemisphere, and humans act as the reservoir hosts in this urban cycle. Acute disease manifests as a febrile illness with polyarthralgia, mainly affecting the joints and causing back pain, headache, fatigue, maculopapular eruption, conjunctivitis, and sometimes gastrointestinal symptoms including vomiting and diarrhea [54]. Chronic complications due to polyarthralgia can cause significant morbidity in some patients [54]. The first autochthonous cases of CHIKV in Brazil were reported in Amapá State in 2014 [55]. Brazil remains the country with the highest number of reported CHIKV cases in the Americas [56,57]. Annual outbreaks are seen in a heterogeneous pattern wherein municipalities without outbreaks the previous year have an increased likelihood of an outbreak in the following year across the country [57].

Recent circulation: Since its introduction into Brazil, CHKV has spread rapidly throughout the country and is now considered endemic with active virus circulation annually [56]. Brazil remains the country with the highest number of reported CHIKV cases in the Americas [57]. Annual outbreaks are seen in a heterogeneous pattern wherein municipalities without outbreaks the previous year have an increased likelihood of an outbreak in the following year across the country [57]. Concurrent outbreaks with dengue virus occur often, making case reporting difficult. Case numbers have significantly increased in the past three years, with the highest case counts occurring in heavily urban areas of the country [58]. One prospective study in São Paulo State evaluated human seroconversion to CHKV over four years found that seroconversion rates increased annually, and although relatively low at the genesis of the study, cryptic virus circulation foreshadows the onset of additional future outbreaks in large populous urban areas like São Paulo [59].

#### 2.1.3. Eastern Equine Encephalitis Virus (EEEV)

Eastern equine encephalitis virus (EEEV) has had a history of causing epizootic encephalomyelitis outbreaks in equines in the Western Hemisphere since the 1930s, and probably decades before, but was unrecognized then. The virus was finally isolated from a febrile horse from a widespread epizootic outbreak in 1933 impacting multiple northeastern states in the USA [60,61]. Although EEEV is the sole species in the EEE antigenic complex, multiple lineages or strains have been identified with molecular, epidemiological, and ecological differences [3]. Three lineages have been identified as South American strains, and one conserved lineage is considered the North American strain [62]. In general, North American infections lead to high virulence in humans and equines compared to the high virulence and attenuated disease in humans in South American strains [63]. Although much is known regarding the transmission cycles of North American strains of EEEV, definitive information on reservoir and amplification hosts and vector species for South American strains is lacking [63,64]. It has been suggested that wild rodents and wild birds play roles as amplification hosts in South America for this virus [65,66]. Epizootics in Central and South America have been primarily reported in equine populations, thought to be transmitted by *Culex* spp. mosquitoes, with limited disease in humans [63,67]. The lack of reported human cases is attributed to presumed differences in the pathogenicity of the South American EEEV strains compared to the North American strains [63,67].

In Brazil, three strains of EEEV have been isolated in the country, most notably associated with wide-scale, fatal, equine epizootics in multiple regions since the 1940s [64,66,68]. A study identified viral RNA within 18 horses across six Brazilian states as recently as 2009 [69]. Although Brazilian *Ae. aegypti* and *Ae. albopictus* mosquitoes from Mato Grosso State have been reported positive for the virus, *Culex* spp., specifically within the *Melanoconion* subgenus are considered the most likely candidates for epizootic and potential human EEEV within the Amazon basin [51,64]. Additional mosquito species from which EEEV has been isolated in Brazil include *Cx. (Mel.) taeniopus*, *Cx. (Mel.) pedroi*, *Cx. (Mel.) spissipes*, *Cx. quiquefasciatus*, and *Ae. taeniorynchus* [41,51]. Equine serosurveys conducted within the country have found equines with up to 47.7% seropositivity (within Mato Grosso do Sul State) among unvaccinated populations [70] and within Pará State [71]. Several large arbovirus surveillance studies between the 1950s and 1980s have isolated EEEV and found antibodies to this virus within wild bird populations, opossums, and wild rodents in Pará State [41,45]. A recent serological study found additional opossums with antibodies to EEEV within Pará State [72]. Human disease in North America can present as a serious febrile and encephalitic disease with case fatality rates up to 50%; incapacitating neurologic sequelae are seen in survivors for years afterwards [63]. Only two individuals have been diagnosed with EEEV causing a febrile illness in Brazil: one from a northeastern region in the 1950s [73] and the second in Mato Grosso State in 2015–2016 [74]. Human serosurveys have suggested that humans living near horse epizootics are at increased risk of infection [66]. Multiple serosurveys of residents living near the Amazon basin in Pará State reported between 0.4 and 4.1% positivity, suggesting that the virus has been circulating for decades [45,75]. A large serosurvey from the 1980s found 5.6% of 516 tested individuals with antibodies to EEEV in São Paulo State [48]. While limited human cases have occurred in the country, EEEV is considered endemic in Brazil.

Recent circulation: Very little information exists on current EEEV circulation within Brazil. The most recent observations include serological evidence from opossums in 2021 [72] and humans in 2021 [66].

#### 2.1.4. Western Equine Encephalitis Virus (WEEV)

Western equine encephalitis virus (WEEV) was originally isolated from a horse that had died from encephalitis in 1930 in California, USA [76]. Since its isolation, it has been attributed to multiple sporadic outbreaks in the western and midwestern USA, and it eventually travelled southward into South America [77]. The enzootic transmission cycle of WEEV is well studied in the USA, where the primary vector, *Cx. tarsalis*, circulates the virus between the amplifying hosts—wild passerine birds and multiple mammals participate in a secondary cycle; humans and equines are considered dead-end hosts [77,78]. In South America, wild birds also play an important role in the enzootic disease transmission cycle, with additional viral activity found in sentinel rodents and disease found in humans and equines. In Brazil, WEEV has been historically found in multiple *Culex* spp. mosquitoes and *Aedes fulvus* along with multiple passerine wild birds [35,79]. It was first isolated in Brazil from wild birds in the 1950s–1960s [41,80]. This virus is also thought to contribute to equine morbidity and mortality in the country along with other equine encephalitic viruses [81]. One serosurvey conducted in Mato Grosso do Sul found approximately 36% WEEV seropositivity in unvaccinated horses [70]. One opossum was found with antibodies to WEEV within Pará State from a large serosurvey in the 1980s [45]. Disease in humans presents as an encephalitic illness with nonspecific symptoms including fever, chills, malaise, and weakness; most symptomatic individuals will recover within a few days without sequelae [42,82]. Neurologic disease progression includes symptoms of headache, confusion, coma, and seizures; fatigue, headaches, and irritability are common long-term symptoms reported in patients who recover from acute neurologic disease [82]. This neurologic form of disease is most common in children and older adults with suppressed immune systems. Although WEEV has been historically found circulating in mosquitoes, birds, and equines in Brazil, human disease has not been reported to date. However, serological evidence of WEEV in Brazil has been reported in humans in Pará and São Paulo States [45,48,75,79], but there is relatively less research and information regarding WEEV human disease in South America compared to the USA.

Recent circulation: There has been no recent literature on WEEV circulation within Brazil through either serological or molecular detection in humans, animals, or mosquitoes.

#### 2.1.5. Venezuelan Equine Encephalitis Virus (VEEV)

Venezuelan equine encephalitis virus (VEEV) is a part of the larger VEE complex, where over 14 subtypes and varieties have been described [83]. The first isolation of the virus was from the brain of an encephalitic horse in 1938 following its death in the Guajira Peninsula of Venezuela [84,85]. Although the first isolation of the virus was in 1938, this virus has been associated with serious equid disease since the 1920s in Latin and South America [83]. The further identification of these virus species has led to the classification of subtype I varieties (AB-F) and subtypes II-VI. Subtype I varieties AB and C are responsible for major equine epizootics and epidemics throughout South and Central America whereas subtype I varieties D-F and subtypes II-VI are enzootic, albeit avirulent in equids [83,86]. Subtype IAB and IC species are maintained in a zoonotic transmission cycle between equids and mosquitoes in agricultural settings; horses, donkeys, and mules have been identified as key amplification hosts for large epizootic outbreaks [83,87]. Culicids incriminated as vectors for these strains of VEEV include *Aedes sollicitans*, *Ae. taeniorhynchus*, *Psorophora columbiae*, *Ps. confinnis*, *Mansonia indubitans*, and *Deinocerites pseudes* [83,88,89,90,91]. The transmission cycles of enzootic subtypes ID-F and subtypes II-VI are relatively less understood than those of subtypes IAB and IC; rodents are thought to be the primary amplification hosts although VEEV has been isolated in bats and birds as well [43]. These amplification hosts infect mosquitoes within the *Culex* genus as the primary vectors [43,83,92,93,94]. All VEEV subtypes and strains can cause instances of febrile illness in humans that are indistinguishable from one another and are commonly misdiagnosed as dengue or other arboviral diseases [83]. Common symptoms include fever, headache, retro-orbital pain, nausea, and vomiting, and a small proportion of infected individuals will develop serious neurological disease with convulsions, disorientation, and sometimes death [86,95]. Post-infection neurologic sequelae have been observed in children [87,95,96]. Hemorrhagic disease has been reported in patients infected with VEEV subtype ID [97,98].

In Brazil, multiple subtypes of VEEV have been reported. Before these subtypes were distinguished, a previous serosurvey from 1953 in Pará State found that approximately 17% of residents tested seropositive for VEEV [75]. Another larger virus investigation study (from 1954 to 1959) within the tropical rainforest of Pará State identified VEEV isolated from sentinel monkeys and rodents; wild rodents; pools of *Hemagogus* spp., *Aedes serratus*, and *Sabethes* spp. mosquitoes; and six human cases, five of which exhibited febrile illness [71]. Both of these studies likely identified one of the now known subtypes described below.

Venezuelan equine encephalitis subtype IF, named the Mosso das Pedras virus, has been isolated from *Culex* mosquitoes in the Ribera Valley, São Paulo State [43,99]. Additionally, antibodies against VEEV serotype IF have been found in residents who live near forested areas in the São Paulo State [43]. Only one outbreak of locally acquired human disease has been documented from the Mosso das Pedras virus to date, where 20/25 soldiers training in the Ribeira Valley were infected with a febrile illness in the 1980s, confirmed through serology [43]. Additionally, one isolate of the Mosso das Pedras virus was obtained from the Belém region from a laboratory-acquired infection in 1987 [100].

Recent circulation: There has been no recent literature on VEEV subtype IF circulation within Brazil through either serological or molecular detection in humans, animals, or mosquitoes.

#### 2.1.6. Mucambo Virus (MUCV)

Mucambo virus (MUCV) is another species within the VEEV complex identified in Brazil (VEEV subtype IIIA)). Originally isolated in the Oriboca Forest in Amazonas State from sentinel capuchin monkeys in 1954, this VEEV subtype is believed to circulate within sylvatic rodents and *Culex* spp. mosquitoes, particularly *Cx. (Mel.) portesi*, which acts as the primary vector [43,101]. It has been isolated from the States of Pará and São Paulo, found through human serology; isolation from febrile human cases; isolation from wild birds, rodents, and opossums, as well as from sentinel monkeys and rodents; and isolation from additional mosquito species including *Ae. serratus*, *Ae. horatory*, *Uranotaenia geometrica*, *Haemagogus* spp., *Mansonia* spp., *Sabethes* spp., and *Cq. venezuelnsis* [41,45,48,80,101,102,103,104]. Although not thought to circulate widely in equines, one study has demonstrated horses presenting with antibodies to MUCV in Pará, Brazil [105]. Other serological studies have documented a wide range of human antibodies to MUCV in Para State: from 1.2% positivity in 1380 humans tested to over 57% of 632 humans tested [41,45,104]. One serological study in the 1980s found 6.6% of 516 tested individuals to have antibodies to MUCV in São Paulo State [48]. Disease in humans is typically characterized by the sudden onset of a mild fever, headache, malaise, and weakness lasting up to 3 days [43].

Recent circulation: Evidence of MUCV circulation was most recently published regarding molecular detection within mosquitoes and vertebrate animals in 2019 [101], and evidence of human infection was circulated through a serological study report published in 2022 [104].

#### 2.1.7. Pixuna Virus (PIXV)

Pixuna virus (PIXV), VEEV subtype IV, is another strain of VEEV found in Brazil. This virus was originally isolated from *Trichoprosopon digitatum* mosquitoes in Belem, Pará, Brazil in 1961 [106]. A second isolation from *Anopheles nimbus* mosquitoes was documented the same year along with isolates obtained from wild rodents in the same region of Pará, Brazil [41,43,80,106]. Although the primary vector and vertebrate hosts are unknown, it is suspected that *Culex* spp., *Aedes* spp., and/or *Psorophora* spp. circulate PIXV among rodents in endemic areas [83,107,108]. Pixuna virus has been isolated from humans and from positive serology screenings from residents in the same area of Pará, Brazil [80,102]; the disease presents as a moderate fever, intense muscle pain, and anorexia for three days [80].

Recent circulation: There has been no recent literature on PIXV circulation within Brazil through either serological or molecular detection in humans, animals, or mosquitoes.

#### 2.1.8. Una Virus (UNAV)

Una virus (UNAV), the closest genetic relative of MAYV, is significantly understudied in South America [35,109]. This virus was originally isolated from *Psorophora ferox* mosquitoes in Belem, Pará, Brazil in 1959 [30]. Although UNAV has been isolated from multiple mosquito species within the Brazilian Amazon since this time, including from *An. nimbus*, *Ps. albipes*, *Ps. lutzii*, *Ae. serratus*, *Ae. fulvus*, and *Ae. leucocelaenus*, *Cq. arribalzaga*, *Culex* spp., and *Wyeomyia* spp., the primary vector has not been determined [35,41,109,110]. Additionally, the transmission cycle of this virus is virtually unknown, with limited information on vertebrates wherein antibodies to UNAV have been identified in wild birds, horses, and humans [35,111,112,113,114], with the majority of these studies coming from Argentina. It is suggested that a rodent reservoir host is the most likely source of virus circulation in the environment [109]. Human disease and symptomology have not been determined for this virus to date.

Recent circulation: There has been no recent literature on UNAV circulation within Brazil through either serological or molecular detection in humans, animals, or mosquitoes.

#### 2.1.9. Aura Virus (AURAV)

Aura virus (AURAV), another highly understudied alphavirus, was originally isolated from *Culex (Melanoconion)* spp. mosquitoes in Belem, Pará, Brazil in 1959 [110]. This virus was then isolated from *Ae. serratus* mosquitoes multiple times over multiple years later and again from additional *Cx. Melanoconion* mosquitoes in the same area [41,110]. Aura virus is thought to be closely related to the Western equine encephalitis and Sindbis viruses [115]. The primary vertebrate host of AURAV is also unknown, but sera from one marsupial, one wild rodent, and one horse each have been reported positive for antibodies to this virus from the same region of Pará [116]. Although there has been no evidence for human disease to date, there is limited evidence of human infection, found through a serological survey, albeit with very low positivity: of one-thousand-three-hundred-and-thirteen residents surveyed in the same forested area in Pará, one positive human was reported [116]. With the lack of information regarding infection, disease, and the transmission of AURAV, this virus is considered potentially non-pathogenic to humans [117].

Recent circulation: There has been no recent literature on AURAV circulation within Brazil through either serological or molecular detection in humans, animals, or mosquitoes.

### 2.2. Flaviviridae

The Flaviviridae family is another important group of viruses for global public health. Twelve viruses of this family have been isolated within the Brazilian Amazon and associated with humans and mosquitoes, multiple of which continue to cause large-scale epidemics annually with notable morbidity and mortality. Flaviviruses are positive-sense RNA viruses; these are generally segregated into three transmission groups: tick-borne, mosquito-borne, and ‘no known arthropod vector’ viruses [118]. A general depiction of the *Flaviridae* transmission cycles and vertebrate animal associations is shown in Figure 2.

#### 2.2.1. Dengue Virus (DENV)

Dengue virus (DENV) is one of the most important arboviruses circulating within the Brazilian Amazon; this virus is responsible for an estimated 100–400 million cases and tens of thousands of deaths annually around the world [2]. Four strains of DENV have been identified globally, and all four have been found circulating within Brazil. Although the virus serotypes had not been isolated yet, there are records of febrile outbreaks of disease throughout Brazil from as early as 1846 and subsequent years in which dengue fever is now suspected as the cause [119]. Dengue virus serotype 1, DENV-1, was originally from a febrile human male who fell ill during an outbreak in the Nagasaki–Sasebo area of Japan in 1942 [120]. After its arrival and subsequent circulation within the Americas in 1977, multiple countries sustained the near-continuous reporting of DENV-1 cases with large epidemics, and these are still occurring today [121]. Dengue virus serotype 2, DENV-2, was originally isolated from male US soldiers stationed in New Guinea in 1944 where several epidemic foci were suspected with up to 27,000 infections but no severe disease [122,123]. This serotype entered the Western Hemisphere as early as 1953 in Trinidad and Tobago, with the frequent reporting of large-scale epidemics since the 1970s [121]. Dengue virus serotype 3, DENV-3, was isolated from both severely febrile and hemorrhagic patients and wild caught *Ae. aegypti* mosquitoes from the Philippines in 1954 [124]. Serotype 3 was initially detected in the Americas in 1963 in Puerto Rico; however, it was not seen widely in the Western Hemisphere until 1994, after a second introduction of a different genotype of DENV-3, leading to wide-scale epidemics and large numbers of dengue hemorrhagic fever cases [125]. Dengue virus serotype 4, DENV-4, was isolated and identified concurrently from *Ae. aegypti* mosquitoes and febrile patients at the same location in the Philippines as DENV-3 in 1954 [124]. This serotype circulated within Southeast Asia for decades before finally reaching the Americas in 1981, where it was first reported in Brazil, Cuba, Dominica, Puerto Rico, and the US Virgin Islands [121].

All DENV serotypes are transmitted primarily by *Aedes aegypti* in the Americas; however, both *Ae. aegypti* and *Ae. albopictus* play important roles in epidemic spread in Southeast Asia [126,127]. Additionally, both *Aedes aegypti* and *Aedes albopictus* in various regions of the world have demonstrated the vertical transmission of DENV, wherein offspring of infected female mosquitoes are infected with the same DENV, independent of taking a blood meal from an infectious host [128]. Various other genera of mosquitoes have been found positive for dengue virus; however, these are not considered vectors. Humans are the known reservoir hosts for all dengue virus serotypes maintaining epidemic cycles of the virus, but it is well known that non-human primates are sylvatic reservoir hosts as all serotypes of the virus have been isolated from various non-human primate species along with positive serology [129]. Additionally, multiple vertebrate mammals have been reported infected with dengue virus, although their respective host status has not been confirmed [129]. Across the globe, dengue virus nucleic acid (through PCR testing) has been reported in monkeys, bats, rodents, marsupials, and dogs, and DENV has been positively identified in monkeys, bats, pigs, rodents, dogs, bovids, horses, marsupials, birds, sloths, armadillos, and shrews through serology [129]. Human disease causes relatively significant morbidity and mortality. Most infected patients who exhibit clinical symptoms experience a sudden onset of fever in combination with myalgia, arthralgia, anorexia, headaches, and severe bone or joint pain [130]. There is no difference in clinical manifestation between the four serotypes; however, the risks for severe disease such as dengue hemorrhagic fever or dengue shock syndrome are significantly increased following a secondary infection by a separate strain of the virus [130].

Since the early 2000s, all four DENV serotypes have co-circulated throughout Brazil to varying degrees of heterogeneity dependent on the seasonality, human population and movement, and presence and re-emergence of *Ae. aegypti* [119,131]. The dengue-1 serotype was introduced into Brazil in 1981 in Roraima State [121,131]. The dengue-2 serotype was first reported in Brazil in Rio de Janeiro State in 1990; both DENV-1 and DENV-2 co-circulated throughout the country, successfully causing multiple large-scale outbreaks with significant cases of dengue hemorrhagic fever for over 10 years [121,132]. Although the exact location and time is unknown for DENV-3′s introduction into Brazil, it is estimated that this serotype entered the northern portion of the country near the Brazilian Amazon, introduced either in Amapá, Pará, Roraima, or Tocantins State in 1999 [131]. Dengue serotype 4 was first introduced into Roraima State in 1981; not much was known regarding its circulation outside of this region until 2008, when DENV-4 was first reported outside of Roraima in Amazonas State and subsequently spread throughout the country in subsequent years [131]. The epidemiology of DENV within Brazil can be described in two distinct periods: between 1980 and 1990, cases did not exceed 100,000; since 1991, DENV cases have gradually increased annually, now reaching more than 1 million annually, with notable outbreaks every 2–3 years due to the concomitant circulation of all four serotypes in the country [131]. Serological studies routinely find a large proportion of tested populations with antibodies to DENV (all serotypes) throughout much of the country [133].

*Aedes aegypti* remains the most important vector for DENV within Brazil, being implicated as the primary vector of the majority of the country’s outbreaks [43,134]. Notably, additional species of mosquitoes have been found infected with DENV, possibly suggesting the silent maintenance of the virus in some regions or additional secondary transmission in highly populated areas. In northeastern Brazil, DENV-1 has been isolated from forest-dwelling *Haemagogus leucocelaenus* mosquitoes [132]. In Mato Grosso State, DENV-4 was isolated from *Culex quinquefasciatus*, *Cx. bidens/Cx. interfor Psorophora varipes/Ps. albigenu*, and *Sabethes chloropterus* [135]. Although *Aedes albopictus*’s involvement in DENV transmission is considered minor in comparison, it is thought that the proportion of transmission events has increased from *Ae. albopictus*, specifically in multiple large-scale DENV outbreaks along the Brazilian coast [132]. Despite the species not formally being implicated in an outbreak, DENV-3 was found in the larvae of *Ae. albopictus* mosquitoes outside of Sao Paulo in the early 2000s, further supporting the role of this *Aedes* species in potential human disease outbreaks and the transovarial transmission of the virus [132]. Additionally, evidence of DENV-1–4 serotypes (through serology only) circulating in non-human primates has been published throughout the entire country, and positive serology for all four serotypes from sloths in Bahia State and horses in Mato Grosso do Sul State has been reported [129].

Recent circulation: Dengue virus is endemic to Brazil and humans, animals, and mosquitoes are positive for virus circulation annually throughout the country. This has been evidenced through serological and molecular studies and annual outbreaks.

#### 2.2.2. Yellow Fever Virus (YFV)

Yellow fever virus (YFV) is thought to be responsible for human disease outbreaks with large-scale morbidity and mortality since the 1600s [136]. However, YFV was not isolated until 1927; the virus was filtered and isolated from a febrile male patient in Ghana, and his blood was also used to demonstrate infection in additional non-human primates [137,138,139]. Although not confirmed, genetic and epidemiological studies hypothesize that YFV was introduced to the Western Hemisphere from endemic African countries through the slave trade in the 16th century, resulting in large-scale outbreaks from the 17th to the early 20th centuries [136,140]. Three transmission cycles are classically described for YFV: sylvatic, rural, and urban. In the sylvatic cycles, zoophilic tree-dwelling mosquito species transmit the virus between non-human primates; in South America, these mosquitoes include multiple *Haemagogus* and *Sabethes* spp. [43,141]. The rural transmission cycle is typically seen in the African savannah region, where semi-domestic mosquitoes that live in both sylvatic and peri-domestic areas transmit the virus between both non-human primates and sometimes humans [136]. Mosquitoes identified in Africa in this cycle include *Ae. aegypti*, *Ae. furcifer*, *Ae. vittatus*, *Ae. bromeliae*, and *Ae. keniensis* [141]. The question of whether the rural cycle exists in South America has yet to be validated or understood [141]. Finally, the urban transmission cycle is perpetuated by *Ae. aegypti* globally; *Ae. albopictus* is hypothesized to play a minor role in transmission as well [141]. Urban yellow fever epidemics occur after a human enters a sylvatic or rural transmission habitat, encounters and is bitten by infected sylvatic or rural vectors, and travels back to an urban environment where they are bitten by urban *Ae. aegypti* mosquitoes. Since the mosquito eradication campaigns of the 20th century organized by the Rockefeller Foundation and the Pan American Health Organization, no urban YFV epidemics have occurred in South America, and virus maintenance is sustained through the sylvatic transmission cycle with the disease originating from human encroachment into sylvatic transmission territory [141,142,143]. The majority of those infected with YFV will be asymptomatic, but those with symptoms will experience general flu-like symptoms and malaise; patients will then enter into the ‘remission’ phase of infection, where symptoms may disappear and seroconversion is observed [1,144]. In 15–25% of symptomatic patients, more serious symptoms will re-appear in this toxic phase, including nausea, muscle pain, vomiting, jaundice, multi-organ dysfunction, and hemorrhagic fever [1,144]. Approximately half of the symptomatic patients within the toxic phase will die within 7–10 days [1,136].

Between approximately 1850 and 1900, in Brazil, YFV outbreaks were limited to specific regions of the country considered endemic for sylvatic transmission—especially regions with the construction of train tracks, coffee farms, and European immigration [119]. Mosquitoes in both the *Haemagogus* and *Sabethes* genera are considered the primary sylvatic vectors in Brazil and implicated for recent outbreaks [144]. Yellow fever virus-infected *Ae. albopictus* mosquitoes have been found in forested edges of the southeastern portion of the country, suggesting their potential role in a rural or even urban transmission cycle [145]. Although urban yellow fever has been eliminated, the potential for its transmission remains high since native *Ae. aegypti* and *Ae. albopictus* are competent vectors in Brazil and sylvatic transmission is continuing to spread [145,146]. Non-human primates constitute the only mammalian reservoir host for sylvatic yellow fever; however, some studies in the Amazon or in regions adjacent to the Amazon have found antibodies to YFV in water buffaloes, sloths, anteaters, and some rodents [45,147,148]. Very little is understood about asymptomatic YFV infection in other mammals. Yellow fever disease can also significantly impact non-human primate morbidity and mortality; some studies have reported declines in some non-human primate populations directly related to sylvatic outbreaks in Brazil [144,149,150,151].

Recent circulation: The entire country has been declared endemic for YFV as recent sylvatic outbreaks between 2016 and 2019 have occurred in new regions thanks to a newly identified genetic lineage of the virus [136,152]. These recent outbreaks have caused significant human and non-human primate morbidity and mortality, with cases reaching 2.8 times higher than the cumulative yellow fever cases in the past 36 years [144].

#### 2.2.3. Zika Virus (ZIKV)

Zika virus (ZIKV) is notably one of the most notorious arboviruses of recent years given the pandemic and world-wide scare from approximately 10 years ago. This *Flavivirus* was first isolated from a sentinel rhesus monkey in the Zika forest of Uganda in 1947; three lineages of the virus are hypothesized to have originated in Sub-Saharan Africa: two clusters of African lineage and one Asian lineage that inevitably reached the Western Hemisphere in 2014 [153,154,155]. In its native range in Sub-Saharan Africa, mosquito vectors responsible for transmission maintenance include forest-dwelling *Aedes* spp. such as *Ae. africanus* and *Ae. furcifer* and peri-domestic *Ae. albopictus*; additional isolations have been from various African *Anopheles* spp. including *An. gambiae* s.l. and *An. coustani* and *Mansonia uniformis* [154,155]. Urban transmission is maintained primarily by *Ae. aegypti* and secondarily by *Ae. albopictus* in Africa and Southeast Asia [154]. In South America, both *Ae. aegypti* and *Ae. albopictus* are the primary urban transmission vectors [154]. Sylvatic vector species in South America have exhibited limited evidence of the successful transmission of ZIKV, so an independent enzootic transmission cycle is not expected to be as widespread as those of other viruses such as YFV [156]. The main reservoir hosts for ZIKV are non-human primates within sylvatic areas; however, some rodents, bats, cattle, water buffaloes, goats, sheep, and ducks have tested positive for ZIKV antibodies in some Asian countries [154]. Some non-human primates have exhibited adverse health effects from infection; however, there has been limited evidence of this in the recent literature [157]. Human clinical manifestations are seen in approximately 20% of infected individuals; these closely resemble symptoms of other arboviral infections including fever, myalgia, headache, arthritis, a macular or papular rash, conjunctivitis, retro-orbital pain, anorexia, vomiting, and stomach pain [154,158]. Neurological complications were seen early during the pandemic, with associations first noted between ZIKV infection and Guillain–Barré syndrome (GBS) in Southeast Asia, and eventually, researchers reported an association between infection in pregnant women and adverse health outcomes in neonates including microcephaly and abortion, first seen in French Polynesia in 2013 [155]. Globally, the prevalence of ZIKV-associated GBS is estimated to affect approximately 1.23% of infected persons, and ZIKV-associated microcephaly affects 2.3% of infected pregnant women [159,160].

Of special note is the sexual transmission route of ZIKV; this was the first arbovirus known to be transmitted through direct sexual contact [161]. Although not related to mosquitoes, this discovery changed how ZIKV epidemiology and control measures were implemented during the pandemic.

Although the exact time of introduction is unknown, ZIKV was first reported in northeastern Brazil (Pernambuco, Rio Grande do Norte, and Bahia States) in 2015 [162,163]. By 2016, ZIKV had been reported in almost every region of the country except for some remote regions in the Amazon and the southernmost portion of Brazil [7]. Following its introduction into the country, during the pandemic, Brazil quickly became one of the most heavily ZIKV-burdened countries in terms of both adult and neonate health impacts and the economic burden [164,165]. Similarly to what is found in both dengue and YF viruses, *Ae. aegypti* is considered the main epidemic vector of ZIKV transmission throughout the country [7,166]. This mosquito species has also shown evidence of transovarial and venereal transmission among conspecifics in Brazil, further complicating the effectiveness of control measures [167,168]. Additionally, ZIKV has been isolated from *Ae. albopictus* mosquitoes in multiple states including Espírito Santo and Bahia during rural outbreaks [169,170] and outside of documented outbreaks within the states of Amazonas, Rio de Janeiro, Sao Paulo, and Mato Grosso [51,171,172,173]. In Pernambuco State, field-collected *Cx. quiquefasciatus* mosquitoes tested positive for ZIKV, and the virus was noted within the salivary glands, suggesting the role of additional mosquito species in transmission of the virus as well [174]. Brazilian non-human primates have tested positive for circulating ZIKV virus through molecular methods; however, the full extent of the effects of this virus on non-human primate populations in sylvatic environments is unknown despite the large diversity of non-human primates in the country and potential for adverse health impacts [154,157].

Recent circulation: Zika virus has been considered endemic throughout the country of Brazil following the 2015–2016 pandemic. Relatively large proportions of tested human populations in Brazil have antibodies to ZIKV, indicating how widespread the virus is [133]. Additional evidence of continued circulation throughout the country is documented through various mosquito surveillance studies in which ZIKV is found in *Aedes* mosquitoes regularly [167,168,169,170,171,172,173].

#### 2.2.4. Ilhéus Virus (ILHV)

Ilhéus virus (ILHV) is a native *Flavivirus* to Brazil; it was first isolated from a mixed pool of both *Aedes* and *Psorophora* mosquitoes, including *Ae. serratus* and *Ps. ferox*, in the vicinity of Ilhéus in the State of Baía in 1944 [175]. This virus is most closely related to Rocio virus, another arbovirus found in Brazil described later in this text, and Japanese encephalitis virus, a *Flavivirus* native to Japan and other Asian counties and the western Pacific [43]. Since its isolation from Brazilian mosquitoes, it has been isolated from mosquitoes, some animals, and humans in multiple countries throughout South and Central America [176,177,178,179,180,181,182,183,184]. Ilhéus virus is thought to circulate within a sylvatic cycle maintained by *Ps. ferox* mosquitoes as the primary vector and additional *Psorophora* and *Aedes* mosquitoes acting as secondary vectors [43,119]. Additional mosquitoes in which ILHV has been isolated from include *Ps. albipes*, *Ps. lutzii*, *Ae. serratus*, *Ae. fulvus*, *Ae. scapularis*, *Ae. leucocelaenus*, *Ae. argyrothorax*, *Ae. sexlineatus*, *Culex coronator*, *Cx. (Mel.) portesi*, *Haemagogus leucocelaenus*, *Ha. spegazzinii*, *Sabethes chloropterus*, and unidentified *Trichoprosopon* spp. [35,41,43,101,118]. Primary vertebrate reservoir hosts include several species of wild birds, and the virus has been isolated from bats and sentinel monkeys; serological surveys have shown antibodies to ILHV in bats, rodents, horses, opossums, and water buffaloes as well [43,72,81,101,118,119]. Ilhéus virus is not associated with epidemic disease to date, only causing sporadic cases of disease [119]. Clinical symptoms present as a mild febrile illness with high fever, severe headache, chills, myalgia, photophobia, and general weakness; patients are symptomatic for 3–5 days [43,184,185]. Considered rare, encephalitic symptoms have been seen and have led to at least one fatality in Brazil; however, the causal origin of this patient’s encephalitic symptoms are under debate [43,184,186]. Disease is often mis-attributed to other arboviral infections such as dengue, St. Louis encephalitis, or even yellow fever [119,184].

In Brazil, over 41 strains of ILHV have been isolated from humans, animals, and mosquitoes; serological studies have revealed ILHV antibody rates as some of the highest within the Brazilian Amazon following those of dengue and yellow fever viruses [43]. Seroprevalence in humans to ILHV has been documented at over 36% in the States of Bahia, Mato Grosso, Pará, and up to 51.3% in Amazonas State; a 1980s serosurvey found just 5.2% of 516 tested individuals to have antibodies to ILHV in São Paulo State [41,45,48,75,133,184]. The virus has been isolated from humans and detected through serological tests from almost every Brazilian state [48,184]. Although human disease cases have not been reported recently, this virus has been isolated and found through serological testing continuously in mosquitoes and other animals. Mosquito species found infected with ILHV in Brazil include *Ps. ferox*, *Ps. albipes*, *Ae. serratus*, *Ae. fulvus*, *Ae. leucocelaenus*, *Ae. scapularis*, and *Cx. (Mel.) portesi* in the Brazilian Amazon; *Ae. scapularis* in the Brazilian Pantanal (Mato Grosso do Sul State); *Cx. coronator* and mixed *Culex (Melanoconium)* spp. in Mato Grosso do Sul State; and mixed *Culex* spp. in São Paulo State [30,41,101,187,188,189,190]. Multiple populations of healthy equids have tested positive through serology for ILHV antibodies within the states of Pará and Mato Grosso do Sul [191], and some equines with neurological disease within Mato Grosso do Sul have tested positive for ILHS antibodies along with multiple other flaviviruses [192]. Although serological evidence has suggested additional mammals infected with ILHV, wild birds are thought to maintain transmission as the primary reservoir hosts in Brazil [81,118,193].

Recent circulation: The most recent evidence of ILHV circulation has been from serological findings on antibodies to the virus within vertebrate animals from 2019 [72,101], a serological study in humans in 2021 [133], and multiple molecular detection studies in mosquitoes in 2019 [189], 2020 [188], and 2022 [190]. Lastly, a fatal case of disease attributed to ILHV was reported in 2020 [186].

#### 2.2.5. West Nile Virus (WNV)

West Nile virus (WNV) is another notable *Flavivirus* due to its rapid global spread and impact on public health systems, mosquito surveillance and control, and medical preparedness for arboviruses—especially in the United States (U.S.) [194]. West Nile virus was originally isolated from the blood of a febrile woman in Uganda in 1937; additional isolations were obtained 13 years later in Egypt [195,196]. The virus eventually spread throughout much of the Middle East, Europe, and Asia, eventually reaching Europe, with noted increases in outbreak frequency in humans and equines, severe human disease, and avian mortality since the mid-1990s [197,198]. West Nile virus was introduced into the U.S. in 1999, spread across the contiguous USA within 4 years, and eventually reached South America in 2004 [199]. Although WNV’s presence has not been confirmed in all South American countries, it is thought that the virus may be circulating within competent vectors and reservoir hosts throughout the continent [199,200]. Mosquitoes within the *Culex* genus are the primary vector species for the transmission of WNV: those that are ornithophilic maintain enzootic transmission while those mosquitoes that feed on birds and humans and other mammals (like horses) propagate epidemics and epizootics [201,202]. In its native range, WNV is transmitted by *Culex univittatus*, *Cx. pipiens*, *Cx. Quiquefasciatus*, and *Cx. theileri* [203,204,205,206,207,208,209]. Notably, mosquitoes within the *Cx. pipiens* and *Cx. quiquefasciatus* complexes are globally important vectors for this virus [197,203,210,211,212]. There is limited information on WNV transmission within South America; however, both *Cx. pipiens* and *Cx. quiquefasciatus* are considered the most important vectors within this region, with additional transmission attributed to *Cx. nigripalpus* as well [3,211,213,214]. Birds within multiple orders have been identified as reservoir and amplification hosts for WNV, including Accipitriformes, Anseriformes, Charadriiformes, Columbiformes, Falconiformes, Galliformes, Passeriformes, Piciformes, Psittaciformes, and Strigiformes [215]. Of note are the orders Columbiformes and Passeriformes, some members of which have been dubbed ‘super-spreaders’, largely responsible for WNV’s expansive and rapid reach across the Western Hemisphere [200,216,217,218]. Research has demonstrated that over 30 other vertebrate animals can become infected with WNV but do not contribute to the enzootic cycle of transmission [219]. Birds within the orders Falconiformes, Accipitriformes, and Strigiformes have been reported with notable clinical manifestations; equines have demonstrated significant disease—with up to 50% case fatality—if infected as well [220,221,222,223]. Human disease is seen in approximately 20% of infected patients, with symptoms presenting as flu-like febrile disease, headache, generalized weakness, muscle pain, and some joint pain—characterized as WNV fever [224,225]. Approximately 1% of WNV-infected patients will develop neuroinvasive disease (not seen in those with WNV fever); the risk for WNV neuroinvasive disease increases with age and in those who are immunocompromised [224,226]. Neuroinvasive disease is classified within three categories: WNV meningitis, WNV encephalitis, and WNV acute flaccid paralysis [225,226]. The case fatality for WNV neuroinvasive disease is approximately 10% [225].

Recent circulation: In Brazil, WNV was not isolated from collected mosquitoes until 2017 despite positive serological findings from humans and various animals throughout the country since 2003 [227]. This first isolation was from a pool of *Culex (Melanoconion)* spp. mosquitoes from the State of Pará [227]. An additional study experimentally infected *Cx. quinquefasciatus* mosquitoes from the Brazilian Amazon with WNV and demonstrated successful transmission in the laboratory [214]. However, the exact species of mosquitoes implicated in natural WNV transmission maintenance within Brazil have not been identified to date. Since its introduction into Brazil, surveillance has been conducted through serology and molecular testing from animals and humans. Antibodies to WNV have been reported from both febrile and afebrile equines in the States of Mato Grosso, Mato Grosso do Sul, Paraíba, Rio de Janeiro, Bahia, and Ceará [192,228,229,230,231,232,233,234,235]. Antibodies to WNV have also been reported from wild and domestic birds in the States of Rio Grande do Sul, Mato Grosso, and Ceará [228,231,232,236]. The virus was first isolated in Brazil from a febrile horse from the State of Espírito Santo in 2018, with subsequent horse deaths associated with the encephalitic disease outbreak in these animals [237,238] and the additional molecular identification of WNV from horses in the States of Espírito Santo, São Paulo, Piaui, Minas Gerais, and Ceará occurring between 2018 and 2021 [232,239,240]. Large scale surveillance efforts have also attempted to find WNV circulating in humans in the country with limited success despite the serological and molecular evidence of the virus circulating [241]. A recent 2019 study found that 2.6% of humans tested for WNV antibodies were positive following a fatal horse encephalitis case [232]. However, a 2021 study found 46.6% of tested individuals positive for WNV antibodies in Amazonas State [133].

#### 2.2.6. St. Louis Encephalitis Virus (SLEV)

St. Louis encephalitis virus (SLEV) is a *Flavivirus* closely related to the West Nile and Japanese encephalitis viruses. This virus was originally isolated from brain tissues of deceased victims of a relatively large outbreak of fatal encephalitis in St. Louis, Missouri, USA in 1933 [242,243]. Prior to the introduction of WNV into the Western Hemisphere, SLEV was the most widely distributed encephalitic arbovirus in this region [43,244]. St. Louis encephalitis virus has most likely existed in the Western Hemisphere—at least within North America—for thousands of years; phylogenetic analyses have not resolved virus movement within the Western Hemisphere completely; however, seven lineages have been described, including two associated with South America [245]. The transmission cycle of SLEV has been extensively studied within the U.S.; however, there is a considerable lack of research into the transmission cycle within South America in comparison [245]. In general, wild birds within the orders Passeriformes and Columbiformes are principal vertebrate reservoir and amplification hosts, particularly within warmer months [43,245]. Mosquitoes in the *Culex* genus transmit SLEV between vertebrate hosts, but there are notable differences geographically between species throughout the U.S. due to the biology, vectorial capacity, and vector competence of regional mosquito species; virulence of SLEV viral strains; and competence of available vertebrate hosts [244,245]. The virus is also thought to be vertically transmitted from female mosquitoes to their progeny and maintained in some bird species during winter months [245]. In South America, rodents and sloths are thought to also play roles as potential secondary reservoir hosts [245]. Mosquitoes within the *Cx. pipiens* complex are considered primary vectors; specifically, *Cx. quiquefasciatus*, *Cx*. *pipiens*, and *Cx. nigripalpus* are consistently implicated in South America [245]. Disease has been described within both equines as well as humans, with potential neurological involvement [246,247]. Human clinical symptoms typically occur in 0.2–7.0% of infected patients; the case fatality rate ranges between 5 and 20%, with an increased risk in the elderly [248,249]. Three syndromes have been described in association with disease due to SLEV infection, with increasing severity: (1) febrile headache with fever, possibly including nausea and vomiting; (2) aseptic meningitis with high fever and stiff neck; and (3) encephalitis with high fever, altered consciousness, and/or neurologic dysfunction [245,249]. In less than 50% of cases, sequelae include fatigue, depression, memory loss, or headache for up to 3 years following initial infection [250,251].

The first isolation of SLEV from Brazil was from *Sabethes belisarioi* mosquitoes from Pará State in 1960 [252]. However, the maintenance of this virus in Brazil is thought to be attributed to *Culex* species including *Cx. coronator*, *Cx. declarator*, *Cx. nigripalpus*, *Cx. pipiens*, and *Cx. quiquefasciatus*, which have been found infected with SLEV in the country [41,43]. Additional mosquito species in Brazil with unknown transmission potential where SLEV has been isolated include *Sa. belisarioi*, *Aedes* spp., *Mansonia* spp., and *Gigantolaelops* spp. mites [35,41]. Epizootics in sentinel monkey populations have also been reported near Belém, in Pará State [80], and in wild monkeys in Mato Grosso do Sul [253,254]. Wild birds are considered the main SLEV reservoir and/or amplification hosts in the Amazon region and are thought to constitute the main reason this virus is disseminated throughout many states in the country. Over 15 species of wild birds have been documented with antibodies to the virus, with particular emphasis on members of the families Formicariidae, Pipridae, and Columbidae [41,43,80,190]. SLEV has also been isolated from wild birds within Pará State [45]. Antibodies to SLEV have been documented in sentinel rodents and opossums in Pará State [41]. This virus has also been detected through serological and molecular methods within healthy populations of equids throughout the country in Minas Gerais (molecular and serological methods), Pará (serological only), Amapá (serological only), Acre (serological only), Paraiba (serological only), Rio de Janeiro (serological only), Sao Paulo (serological only), Rio Grande do Sul (serological only), and Mato Grosso do Sul (serological only) States [70,191,233,234,246,253,255]. Neurological disease has also been reported from equines found infected with SLEV in Mato Grosso do Sul [192,247]. This virus has also been reported in large populations of water buffaloes throughout the State of Pará, with no evidence of disease in these ruminants [148]. Despite its proclivity to cause significant outbreaks in North America, large human outbreaks of SLEV have not been documented in South America, and the understanding of SLEV epidemiology within humans is lacking in Brazil [244,246]. Two human cases were reported in Para State in 1980, and the virus was first isolated from the country from one of these patients [256]. The third SLEV human case was reported from São Paulo State in 2004 [257,258], and a small outbreak of the virus was reported in the same region during a dengue virus outbreak in 2006 [259]. Antibodies to SLEV have been detected in humans throughout the Amazon basin since 1953 (initially in Pará State); SLEV antibody prevalence is commonly reported in this region at approximately 5% [75]; however, one serosurvey in Ceará State demonstrated prevalence in a select community experiencing a dengue fever outbreak of up to 10.1% [43]. A few serosurveys have detected between 3 and 6.8% positivity for human antibodies to SLEV within São Paulo State from the 1970s to the 1980s [48,260]. Antibodies and the molecular detection of SLEV were also documented in healthy humans within Minas Gerais State from samples collected in 2012–2013; 4.6% of people tested positive for antibodies and 1.25% had SLEV RNA detected from PCR amplification [246]. In Amazonas State, 54.7% of tested individuals had antibodies to SLEV in a recent 2021 study [133]. Human SLEV cases are thought to be commonly misdiagnosed as dengue fever throughout the country, so the actual prevalence of infection in Brazil is likely underreported [244]. A molecular study on the genome of SLEV suggested that two strains of SLEV are circulating within Brazil: genotype VIII within the northern Amazon and genotype V with a wider distribution throughout the country and South America [261].

Recent circulation: The most recent evidence of the circulation of SLEV within vertebrates has been documented through the detection of virus antibodies within sentinel monkeys reported in 2019 [254] and horses in 2019 [246], 2021 [255], and 2022 [234]. Recently, the molecular detection of virus in humans was reported in 2019 [246], and antibodies from tested humans were reported in 2021 [133].

#### 2.2.7. Bussuquara Virus (BSQV)

Bussuquara virus (BSQV) was originally isolated from a sentinel howler monkey in Pará State, Brazil in 1956 [262]. The virus has since been isolated from sentinel rodents, wild rodents, and multiple mosquito species within the same region of the country including *Culex melanoconiun* spp., *Cx. declarator*, *Cx. (Mel.) portesi*, *Cx. (Mel.) taeniopus*, *Cx. (Mel.) pedroi*, *Cq. venezuelensis*, *Mansonia titllans*, and *Sabethes* spp. and from some black flies (*Simuliidae* spp.) [29,30,35,41,71,101]. A relatively recent serological study found evidence of water buffaloes being infected with BSQV in Para State [148]. Although BSQV has been found in various animal and mosquito species, it is hypothesized that primary vectors are within the *Culex* genus, and the virus is primarily circulated within wild rodents as host animals [35]. Bussuquara virus is categorized as a pathogenic arbovirus for humans [42], and one record of human infection has been noted from 1956 [30]. A 1980 serosurvey in São Paulo State found 0.6% of 516 tested individuals to possess antibodies to BSQV [48]. The disease was characterized as febrile with anorexia, joint pain, chills, sweating, and headache [263,264].

Recent circulation: Bussuquara virus has only been found recently within mosquitoes in Brazil, reported in 2019 [101].

#### 2.2.8. Rocio Virus (ROCV)

Rocio virus (ROCV) is a flavivirus closely related to both the Ilhéus and Saint Louis encephalitis viruses. This virus was first isolated from the spinal cord tissues of a fatal case of encephalitis from São Paulo, Brazil in 1975 during widespread outbreaks of meningoencephalitis in multiple local coastal communities [106,265,266]. Over 1000 cases are suspected to have occurred during this initial outbreak [265,266]. The transmission cycle of ROCV is not well understood; however, it is likely maintained between mosquitoes and birds within an enzootic cycle, with humans and equines acting as incidental dead-end hosts [267]. This virus was isolated from *Ps. ferox* mosquitoes months following the original outbreak and viral isolation from encephalitis patients in São Paulo; it is widely suggested that some *Aedes* spp. including *Ae. scapularis* and *Culex* spp. including *Melanoconion portesi* and *Cx. quiquefasciatus* play important roles as bridge vectors [43,101,267,268,269,270,271]. Wild bird species within the Passeriformes order are considered the primary reservoir hosts of ROCV, with migratory birds playing a large role in virus distribution throughout Brazil [43,267]. Domestic birds such as chickens, ducks, and pigeons are epizootic hosts, thought to support bridge transmission to peri-domestic dead-end hosts and humans [267]. Rocio virus antibodies have been detected within the following animals through serological surveys: equines in Rio de Janeiro, Mato Grosso do Sul, Paraiba, São Paulo, and Mato Grosso States; buffaloes in Pará State; and birds, bats, marsupials, and wild rodents in São Paulo State [233,267]. Clinical disease in humans begins with the abrupt onset of fever, headache, anorexia, nausea, vomiting, myalgia, and malaise; coma with respiratory symptoms may appear in severe cases [43,266,267,272]. Encephalitic symptoms may appear later, including weakness, confusion, motor impairment, meningeal irritation, cerebellar syndrome, and seizures [43,267,272]. Neuropsychiatric sequelae including visual, olfactory, and auditory disorders; paresthesia; difficulty swallowing; lack or motor coordination; and memory defects have been seen in approximately 20% of patients [43,266,267,272]. The Rocio virus infection case fatality rate has been reported at 10% [267]. Following the large initial outbreak of ROCV in São Paulo State, sporadic cases have been reported and confirmed through blood and serology in São Paulo State in 1978, 1987, and 1990 [48,273,274,275]; Bahia State in 1984 and 1995 [150,276]; and in Goiás State in 2012 (also confirmed through molecular PCR testing) [277].

Recent circulation: Molecularly confirmed clinical cases were last described in 2012 during a dengue virus outbreak [277]. Additionally, a recent serological survey among military trainees in Amazonas State found 48.7% of the tested individuals to possess antibodies to ROCV although potential cross-reactivity to other flaviviruses could not be ruled out [133].

#### 2.2.9. Cacipacore Virus (CPCV)

Cacipacore virus (CPCV) was originally isolated in 1977 from the whole blood of a black-faced antthrush bird (order: Passeriformes) in Pará State, Brazil [30,106]. Phylogenetically, CPCV shares a common ancestor with multiple other arboviruses found in the region including St. Louis Encephalitis virus, Usutu virus, and West Nile virus [278]. The ecology and transmission cycle of CPCV is largely unknown; however, evidence demonstrates that the virus is maintained in nature within multiple vertebrate hosts from multiple mosquito species, many of which will bite humans and begin a suggested urban cycle of virus circulation [279]. Cacicapore virus was isolated from *Amblyomma cajennense* ticks in São Paulo State—however, it is postulated that the ticks are not competent transmission vectors but were subsequently found positive as the host animal (a capybara) was found with CPCV circulating within its blood [278,279]. Mosquitoes are thought to be the main vectors of CPCV in nature as multiple pools of *Culex*, *Anopheles*, and *Aedes aegypti* mosquitoes from Rodônia and Amazonas States were positive for the virus in a study [280]. Serological studies have documented antibodies to CPCV within the following vertebrate animals: wild birds, rodents, bats, buffaloes, and marsupials in Pará State; equines and non-human primates in Mato Grosso do Sul State; and equines in Amapá, Paraíba, Ceará, Bahia, and Acre States [106,148,191,192,235,281,282,283,284]. Additionally, Cacicapore virus has been detected through genetic PCR tests in rodents (clinically ill capybaras) in São Paulo State [278]. A late 1970s study in Pará State found 2/2500 (0.08%) seroprevalence in humans at the time, and no other studies were successful in finding evidence of human infection until the early 2000s [106,279]. The first isolation of CPCV from a human was reported in 2002 from Rondônia State [285]. This first and only known human case was admitted to the hospital with a suspected yellow fever virus or leptospirosis infection [285]. Given that this patient constituted the only evidence of human infection to date, knowledge on the clinical symptoms of CPCV is limited. This patient had traces of blood in the urine, reduced red blood cell counts and hemoglobin levels, jaundice, renal insufficiency, nausea and vomiting, and diarrhea [279,285]. This index patient passed, and the following post-mortem examination revealed a leptospirosis and CPCV coinfection, and the patient’s death being caused by CPCV infection was speculative [279,285].

Recent circulation: There has been no recent literature on CPCV circulation within Brazil through either serological or molecular detection in humans, animals, or mosquitoes.

### 2.3. Peribunyviridae

A relatively new designated family of viruses, the Peribunyaviridae family, includes the *Orthobunyavirus*, *Herbertvirus*, *Pacuvirus*, and *Shangavirus* genera, although only the *Orthobunyavirus* genus is associated with mosquitoes and human infection in Brazil [286]. Many of these viruses infect multiple types of mammals, including humans [286,287]. The majority of the peribunyaviruses are arthropod-borne, transmitted by midges, sand flies, mosquitoes, or ticks; these are all enveloped, negative-sense RNA viruses [286,288]. A general depiction of the *Peribunyaviridae* transmission cycles and vertebrate animal associations is shown in Figure 2. Please note that unlike in other sections, we have not included sections for each virus on recent vs. historical updates. In general, the majority of arboviruses in the Peribunyaviridae family have not been studied in the twenty-first century, making the scientific literature >25 years old for many of the following viruses.

#### 2.3.1. Tacaiuma Virus (TCMV)

Tacaiuma virus (TCMV) was originally isolated from the blood of a sentinel capuchin monkey within the Oriboca Forest in Pará State, Brazil in 1955 [71]. Since its isolation, TCMV’s presence has been documented outside of Brazil three times: antibodies were found within wild birds and humans in Argentina [43,106], and TCMV was isolated from *Wyeomyia* spp. mosquitoes in Colombia [289,290]. The transmission cycle of TCMV is not well understood; however, it is suggested that *Anopheles* mosquitoes are the main vectors—the virus has been isolated from *An. cruzii* in São Paulo State—with additional transmission by other mosquito species from which the virus has been isolated including *Haemagogus janthinomys*, *Aedes triannulatus*, and *Ae. scapularis* in Amazonas State [30,35,41,43,291,292]. The main amplification host(s) have not been determined; however, a somewhat complicated transmission cycle is suggested. Non-human primates are thought to play an important role in virus maintenance as capuchin monkeys are the only non-human vertebrates from which virus has been isolated [41,292]. The highest levels of antibodies have been reported in bats, wild rodents, and wild birds in Pará, Amapá, and Ceará States, suggesting these animals may play an important role within the transmission cycle [43,292,293]. Additionally, horses and water buffaloes within the States of Pará, Mato Grosso, and Mato Grosso do Sul and horses and small ruminants in Ceará State have been found with antibodies to TCMV, suggesting their exposure and potential importance in the circulation of the virus [81,106,232,294,295]. Clinical disease in humans presents with abrupt fever, headache, chills, myalgia, arthralgia, and weakness; these symptoms can last for between 3 and 5 days [43,296,297]. TCMV was isolated from a human in São Paulo State during a previous ROCV outbreak, and antibodies have been consistently found in rural populations of people within the Brazilian Amazon, Central Brazil, and Northern Argentina in approximately 0.5–1.0% of those tested [43,45,113,298]. Additionally, at least two patients from which TCMV was isolated were also found to be concomitantly infected with *Plasmodium falciparum* malaria from Pará State in the 1980s [299].

#### 2.3.2. Boracéia Virus (BORV)

Boracéia virus (BORV) is closely related to Tacaiuma virus, but within a separate serogroup, *Anopheles B*. This virus was originally isolated from *Anopheles cruzii* mosquitoes from the Casa Grande region of São Paulo State in 1962 [300]. Approximately 12 years later, BORV was isolated from multiple pools of *Wyeomyia (Phoniomyia) pilicauda* and *An. cruzii* mosquitoes within the same region of São Paulo State and was considered the causative agent for an infectious illness within residents of a neighboring village in 1974, and neutralizing antibodies were detected in 23.6% of the population sampled there [301]. Evidence of domestic and wild vertebrate infection was noted in this same outbreak in Casa Grande, with domestic cattle, dogs, equines, chickens, and geese positive for antibodies and some wild rodents, marsupials, and birds being positive for antibodies as well [301]. Little else is known regarding BORV’s transmission cycle and prevalence within Brazil.

#### 2.3.3. Tucunduba Virus (TUCV)

Tucunduba virus (TUCV) was originally isolated in 1955 from *Wyomeyia* mosquitoes within the Oriboca Forest of Pará State, Brazil [43]. Since its original isolation, TUCV has been found in multiple mosquito species within the country. Over 50 strains have been isolated from over 14 mosquito species including *Anopheles* spp. including *An. nimbus*, *Ae. fulvus*, *Ae. scapularis*, *Ae. argyrothorax*, *Ae. serratus*, *Ae. sexlineatus*, and *Ae. seplemstriatus*; *Ps. ferox; Hg. leucoceiaenus*; *Cx. coronator* and *Cx. ocellatus; Limatus flavisetosus* and *Li. durhanii; Wyeomyia* spp. including *Wy. aporonomo; Sabethes* spp. including *Se. quassicyaneus* and *Se. interinedius;* and *Trichoprosopon* spp. including *Tr. digitatum*., with the literature suggesting that *Wyomeyia* spp. mosquitoes play an important role as primary vectors in the environment [35,43]. Despite researchers finding TUCV in various mosquito species throughout the Brazilian Amazon, the vertebrate animal host of this virus remains unknown [43,292]. One isolate of TUCV was identified in sentinel suckling mice in São Paulo State between 1974 and 1981 [302]. However, recent large-scale serosurveys for TUCV testing various animals including caimans, equines, and sheep within the Pantanal region produced negative results [303]. Tucunduba virus has been isolated from one fatal human case: a female child presenting with acute fever and neurological impairment including meningoencephalitis with fever, headache, vomiting, and paresis that progressed to coma and death [43,292,304]. This child was the first reported patient infected with this virus in the late 1980s [304]. Three family members of the fatal patient tested positive for CF antibodies to TUCV at the time of this case [297].

#### 2.3.4. Maguari Virus (MAGV)

Maguari virus (MAGV) is closely related to Cache Valley virus (CVV), a mosquito-borne orthobunyavirus rarely reported in humans in the United States of America [305]. Originally thought to be CVV, MAGV has since been classified as a genetically separate virus [305]. This virus was originally isolated from a pool of mixed mosquito species collected from the Utinga Forest within Pará State, Brazil in 1957 [71]. This mixed pool of mosquitoes included *Ae. scapularis*, *Ae. serratus*, *Ae. sexlineatus*, *Mansonia* spp., and *Ps. ferox* [71,306]. A separate large-scale arbovirus investigation in Pará State from the 1950s to the 1960s isolated MAGV separately from *Ae. leucocelaenus*, *An. nimbus*, and *Ps. ferox* mosquitoes [41]. Before its classification as MAGV, an isolate of this virus was also found in *Ae. scapularis* mosquitoes in Trinidad in 1958, thought to be CVV at the time [306,307]. Since its isolation in Pará State, MAGV has also been isolated from various mosquito (*Aedes* spp. including *Anopheles* spp., *Culex* spp., *Wyeomeyia* spp., and *Psorophora* spp.) and animal species in the neighboring South and Central American countries of Trinidad and Tobago, Ecuador, Colombia, Guyana, French Guiana, Argentina, and Peru [106,305,308]. Although this virus has been found widespread throughout South and Central America, much is still unknown regarding its transmission cycle including primary arthropod vectors and animal hosts. In Brazil, MAGV was isolated from sentinel sucking mice from São Paulo State between 1974 and 1998 [302]. Additionally, multiple serosurveys within the Pantanal region (Mato Grosso do Sul and Mato Grosso States) reported antibodies to MAGV within equines (18% and 28% prevalence) and sheep (0.4% prevalence) [81,294,303]. A recent serosurvey within Pará State also identified MAGV antibodies within 7.33% of tested water buffaloes, the highest proportion of viral antibodies of all eight arboviruses investigated in the study [295]. Additional mosquitoes within the Brazilian Amazon from which MAGV has been successfully isolated include *An. nimbus*, *Ae. fulvus*, *Hg. leucocelaenus*, *Mansonia* spp., *Psorophora* spp. including *Ps. albipes*, *Cx.*
*(Mel.) pedroi*, *Limatus* spp., and *Wyeomyia* spp. [35]. Although no human cases of disease have been reported in Brazil, one febrile patient from whom MAGV was isolated from Ucayali Peru in 1998 proved that this virus does cause disease in humans [305]. One large-scale serosurvey within Para State found evidence of human antibodies (~4% of 1398 humans tested) to MUCV from the 2000s [44]. Serological evidence of human infection has also been reported from Argentina, Brazil, Peu, Colombia, and French Guiana, with limited evidence of human disease [106,113,305,309].

#### 2.3.5. Anhembi Virus (AMBV)

Anhembi virus (AMBV) was first isolated from *Wyeomyia pilicauda* mosquitoes in São Paulo State, Brazil in 1965 [310]. Ten years following its initial isolation, AMBV was also isolated from additional pools of both *Wyeomyia pilicauda* and *Trichoprosopon pallidiventer* mosquitoes and, additionally, from a wild spiny rat (*Proechimys iheringi*) from São Paulo State [311]. Although much is unknown regarding the identification of proven arthropod vectors and vertebrate host animals, the literature suggests that *Wyeomyia* mosquitoes are most likely associated with this virus [292]. Serological evidence has also been documented from wild birds captured in São Paulo State without specific details on date and species [311]. One paper has documented evidence of human antibodies to AMBV in Brazil; however, the details of the date of this study are unknown, though this serological evidence has been documented for the Casa Grande region of São Paulo State [30,106,311]. Despite the lack of information on Anhembi virus’s ecology, one study has proposed, through phylogenetic analyses, that this virus’s common ancestor with the closely related arboviruses known as the Guaroa and Wyeomyia viruses was introduced near the Amazon River within Amazonas State around 1764, and AMBV and others subsequently emerged over time within the country [312].

#### 2.3.6. Macauã Virus (MCAV)

Macauã virus (MCAV) was originally isolated from a pool of *Sabethes soperi* mosquitoes in 1976 in Sena Madureira, Acre, Brazil [42,106]. Additional isolations have come from multiple species of wild birds and wild rodents within the Brazilian Amazon [35,292]. Additionally, human serology has been reported positive to antibodies to MCAV within the Amazon; however, the reporting literature has not included the corresponding date and location [30,292]. A member of the Wyeomyia group of viruses, this species is also thought to have emerged in Acre over time after the introduction of its common ancestor with the Guaroa and Wyeomyia viruses in 1764 due to phylogenetic analyses [312]. Much is still unknown regarding the ecology of this virus and its maintenance and transmission cycle.

#### 2.3.7. Guaroa Virus (GROV)

Guaroa virus (GROV) was originally isolated from a woman in Guaroa, eastern Colombia in 1956 [313]. This settlement had had previous recent reported cases of severe febrile illness in multiple residents, although the index patient was not ill at the time [313]. *Anopheles* mosquitoes, especially *An. triannulatus* and *An. nunestovari*, have commonly been found with GROV and are thought to be the primary vectors of this virus [43]. In Colombia, *An. neivai* mosquitoes have been implicated as vectors [314]. Guaroa virus is thought to circulate among avian amplification hosts within its transmission cycle as significantly high levels of antibodies and several viral isolations have been documented in wild birds within the Brazilian Amazon [43,292,293]. Antibodies to GROV have also been detected in water buffaloes in Pará State and sheep in Mato Grosso do Sul and Mato Grosso States within the Pantanal region [295,303]. Disease in humans is characterized as acute high fever, chills, headache, body and joint pain, myalgia, and malaise [43]. Paralysis may also be a severe symptom associated with this virus [315]. Guaroa virus’s orthobunyavirus lineage was thought to have been introduced or emerged in the Brazilian Amazon approximately 250 years ago, and this virus has successfully circulated throughout the region since this time [312]. Additional South American countries with evidence of GROV circulation through serosurveys include Argentina, Peru, Bolivia, and Guatemala; GROV is thought to be widely distributed throughout Central and South America [43,252,316]. Additionally, Guaroa virus isolates have continued to be reported from various human patients throughout South America since its isolation [312]. Several reports of coinfections of GROV with malaria have also been reported within neighboring regions of the Peruvian Amazon and Pará State of Brazil [299,317]. Guaroa virus was first isolated from a human in Brazil in 1964, when several febrile patients presented with disease in Pará State [304]. Additional isolations were made from five men in Pará State were tested in a larger arboviral investigation within the region [41]. Multiple serosurveys have demonstrated relatively high proportions (~1.8–18%) of tested residents with antibodies to GROV within Pará State and 0.6% positivity in São Paulo State [41,44,45,48,71,304,316].

#### 2.3.8. Serra Do Navio Virus (SDNV)

Serra do Navio virus (SDNV) was originally isolated from a singular pool of *Ae. fulvus* mosquitoes in 1966 within Amapá State, Brazil [42,318]. No other record of this virus’s isolation has been documented—from arthropods, animals, or humans. However, serosurveys of local animals demonstrated wild rodents and opossums with evidence of antibodies to SDNV in Amapá State in the 1970s [79,318]. Although no clinical illness has been documented within humans, evidence of infection through multiple serosurveys has shown antibodies (3.4% of 263 tested individuals) present in residents of three separate locations within Amapá State around the same time [79,318]. Much is still unknown regarding the transmission cycle, maintenance, and disease potential of this arbovirus.

#### 2.3.9. Apeu Virus (APEUV)

Apeu virus (APEUV) was originally isolated from the Oriboca Forest in Pará State, Brazil from sentinel capuchin monkeys in 1955 [71]. Mosquitoes within the *Culex* genus and *Melanoconion* subgenus are considered the primary arthropod vectors, with some isolations from some *Aedes* species as well; mosquito species in which APEUV has been isolated include *Cx. (Mel.) portesi*, *Cx. (Mel.) aikenii*, *Ae. Arborealis*, and *Ae. septemstriatus* [30,35,41,43]. Opossums are thought to be the main vertebrate animal amplification hosts for the natural transmission cycle of APEUV; since the specific opossums from which APEUV has been isolated have been arboreal, the literature suggests that APEUV cycles within the forest canopy among arboreal marsupials and non-human primates (New-World monkeys) as secondary amplification hosts [41,292,319,320]. In addition to opossums, the virus has been isolated repeatedly from sentinel and wild monkeys and sentinel rodents close to its original isolation location in Pará State [35,41,71]. Antibodies to APEUV have also been found within wild capuchin monkeys and black howler monkeys in Tocantins State and in sheep from the Brazilian Pantanal Region (Mato Grosso do Sul and Mato Grosso States) [303,321]. Human disease is characterized by a febrile illness including high fever, headache, chills, myalgia, joint pain, photophobia, and retrobulbar pain lasting 4–5 days on average and all patients fully recover [43,71,185,322]. In Brazil, APEUV has been historically isolated from febrile patients within the Brazilian Amazon region (first isolated from humans in 1958)—also within the same region where it was originally isolated, from Pará State [35,41,71,322]. In a large-scale arboviral investigation in Pará State from the 1950s to the 1960s, 15% of tested individuals also had antibodies to APEUV [41]. There has been limited recent literature on human disease and the spread of this virus outside of Brazil; however, APEUV is thought to continuously circulate within the Brazilian Amazon through its sylvatic cycle [292].

#### 2.3.10. Caraparu Virus (CARV)

Caraparu virus (CARV) was originally isolated from a sentinel capuchin monkey from the Utinga Forest within Pará State in 1956 [71]. Similarly to other Group C orthobunyaviruses, the primary vector species are thought to be *Culex* spp. mosquitoes [30,292]. Mosquitoes from which CARV has been isolated include *Cx. (Mel.) aikenii*, *Cx. (Mel.) vomerifer*, *Cx. (Mel.) portesi*, *Cx. (Mel.) sacchettae*, *Cx. (Mel.) spissipes*, *Cx. coronator*, *Cx. nigripalpus*, *Cx. accelerans*, *Cx. (Mel.) amazonensis*, *Wy. medioalbipes*, *Sa. spp.*, *Li. durhamii*, *Ps. ferox*, *Ae. scapularis*, and *Ae. serratus* [35,41,71,292]. Similar to the wide range of associated mosquitoes, multiple species of animals have been found with virus DNA or antibodies to CARV [43,292]. Nocturnal terrestrial vertebrate animals (i.e., wild rodents) are thought to play the most important role as amplification hosts [319]. Since its original isolation and identification, CARV has been isolated from various *Culex* mosquitoes from Peru and Trinidad and isolated from sentinel hamsters in Peru [183,323,324]. Human disease has also been documented from the neighboring country of Peru [325]. Of the Group C orthobunyaviruses, CARV is thought to be the most widely distributed in the Amazon region; evidence of this virus through serology and isolation within animals has been documented within Pará (water buffaloes, capuchin monkeys, bats, and both wild and sentinel rodents), São Paulo (wild birds), and Ceará States (equines) [30,41,43,71,232,292,295,320,326]. The disease is similar to those caused by other Group C viruses, described by a general febrile illness with symptoms such as high fever, chills, headache, myalgia, muscle aches, and malaise [43,71]. In Brazil, human disease cases have been documented and confirmed through molecular testing throughout the country, including within Pará State in the 1950–1960s and São Paulo State in the 1980s [41,43,71,327]. Evidence of human infection through serology (between 0.9 and 15% positive) has also been documented within Bahia, Pará, and São Paulo States [44,45,48,328,329].

#### 2.3.11. Itaqui Virus (ITQV)

Itaqui virus (ITQV) was originally isolated from a sentinel capuchin monkey within Pará State in 1956 [330]. Since its original isolation, ITQV has been isolated from sentinel rodents and *Culex* mosquitoes in both Peru and Venezuela [183,324,331,332]. Potential vectors for ITQV include *Culex* spp. mosquitoes as this virus has been isolated from *Cx. (Melanoconion) spissipes*, *Cx. (Mel.) vomerifer*, *Cx. (Mel.) portesi*, and *Cx. (Mel.) aikenii* within the Brazilian Amazon [35]. One recent vector competency study from Peru found that *Cx. coronator*, *Cx. (Mel.) gnomatos*, *Cx. (Mel.) pedroi*, and *Cx. (Mel.) vomerifer* successfully transmitted ITQV [333]. Wild rodents within the forest are thought to be the primary vertebrate host animals for natural sylvatic transmission [330]. Evidence of viral presence through isolation has been documented from sentinel rodents, sentinel monkeys, opossums, and wild rodents within Pará State [35,330]. Antibodies within rodents and marsupials have been reported from the same region within Pará State as its original isolation [41]. Human disease is characterized by general febrile symptoms such as high fever, chills, headache, myalgia, and malaise, much like with other Group C orthobunyaviruses [43,330]. In Brazil, ITQV has been isolated from humans within Pará State dozens of times since its original isolation, and one serological study from Pará State documented humans (3% of 97 participants tested) with antibodies to this arbovirus in the 1960s [41,330].

#### 2.3.12. Marituba Virus (MTBV)

Marituba virus (MTBV) was isolated originally from a sentinel capuchin monkey within the Oriboca Forest in Pará, Brazil in 1954 [71]. Since its original isolation within Brazil, MTBV was isolated from both sentinel hamsters and *Culex* mosquitoes in Peru; however, there has been little in the literature regarding recent reports of this virus [324]. Like the other Group C orthobunyaviruses, *Culex Melanoconion* mosquitoes are thought to transmit MTBV within forested environments as mosquitoes from this subgenus only have been found positive with MTBV [35,292]. Antibodies to MTBV have been reported from tested marsupials within the Brazilian Amazon while the virus has been isolated from both wild rodents and sentinel monkeys; it is thought that both rodents and marsupials play important roles as vertebrate host animals for this virus in nature [35,43,292]. Symptoms of Marituba virus disease are characterized by high fever, headache, chills, myalgia, photophobia, and dizziness [43,71]. In Brazil, MTBV has been isolated historically from humans within the Oriboca Forest since the 1950s from several febrile patients presenting with disease and positive serological surveys [41,71].

#### 2.3.13. Murucutu Virus (MURV)

Murucutu virus (MURV) was originally isolated from a sentinel capuchin monkey within the Oriboca Forest of Pará State in 1955 [71]. *Culex* mosquitoes are thought to be the primary vector species of interest, with isolations found within *Cx. (Mel.) caudelli*, *Cx. (Mel.) aikenii*, *Cx. (Mel.) portesi*, and *Cx. (Mel.) vomerifer;* MURV has been isolated from *Sabethini* spp. within the same region of its original isolation in Pará State as well [41,71]. A recent vector competency study from Peru indicated that *Cx. coronator*, *Cx. (Mel.) gnomoatos*, *Cx. (Mel.) pedroi*, and *Cx. (Mel.) vomerifer* successfully transmitted MURV [333]. A report from the 1950s to the 1960s indicates that MURV was also isolated from a pool of Ixodid ticks; however, no other information was provided on the species or significance of this finding [41]. Murucutu virus has been isolated multiple times within Pará State from additional sentinel capuchin monkeys and wild rodents [41,71]. Also within the Oriboca Forest region, antibodies to MURV were found within multiple species of wild rodents and marsupials in the 1950s–1960s [41]. Unlike related orthobunyaviruses, MURV has been isolated from wild birds and a singular sloth—both within the Brazilian Amazon, Pará State [41,106,293]. More recently, antibodies to MURV were detected within sheep in the Brazilian Pantanal region, within Mato Grosso do Sul and Mato Grosso States [303]. The disease is similar to those of related orthobunyaviruses, with symptoms including high fever, chills, headache, body aches, dizziness, and photophobia [43,71]. In Brazil, MURV isolates were documented from humans along with human antibodies (4% positive of 97 tested individuals) within the Oriboca Forest as early as the late 1950s and 1960s [41,71,334].

#### 2.3.14. Oriboca Virus (ORIV)

Oriboca virus (ORIV) was isolated from a sentinel capuchin monkey within the Oriboca Forest in Pará State, Brazil in 1954 (interestingly, a strain of VEEV was isolated from this same monkey three days prior) [71]. Oriboca virus has been isolated from multiple pools of *Sabethes*, *Psorophora*, and *Mansonia* spp. of mosquitoes within the same region of its original isolation in the 1950s, indicating that this virus naturally circulates within a forested habitat [71]. Specific mosquito species from which ORIV has been isolated since its original isolation include *Cx. (Mel.) portesi*, *Cx. (Mel.) spissipes*, *Cx. (Mel.) caudelli*, *Ae. arborealis*, *Ae. serratus*, *Ae. argyrothorax*, *Ps. ferox*, *Cq. arribalzagae*, *Cq*, *venezuelensis*, and mixed *Wyeomyia* spp [35,41,335]. Also similar to what applies for other Group C orthobunyaviruses, rodents are thought to be the primary vertebrate animal hosts as ORIV has consistently been found in wild and sentinel rodents and opossums in Pará State, along with consistent reports of antibodies found within these animals—typically near its original isolation location [35,41,43,292]. A more recent serological survey of animals within the Brazilian Pantanal Region, including Mato Grosso do Sul and Mato Grosso States, found a small percentage of sheep to be positive for antibodies to ORIV [303]. A definitive understanding of the transmission cycle of this arbovirus is lacking within the literature. Consistent with other orthobunyaviruses, human disease symptoms include headache, high fever, muscle weakness, nausea, chills, and photophobia [43,71]. In Brazil, ORIV was first isolated from humans within the Oriboca Forest from two laborers in the area [71]. Consistent small serological surveys within areas surrounding the Oriboca Forest in the 1950s–1960s found antibodies to ORIV in up to 25.5% of persons tested [71].

#### 2.3.15. Catu Virus (CATUV)

Catu virus (CATUV) was originally isolated from a febrile human male who was working within the Oriboca Forest, Pará State in 1955 [71]. Since its original discovery, CATUV has been recorded outside of the Brazilian Amazon: there was at least one human disease case in Trinidad, CATUV was isolated from *Culex* spp. mosquitoes in Surinam, and CATUV was isolated from both *Cx. Melanoconion* mosquitoes and opossums in French Guiana [336,337,338,339,340]. The primary arthropod vector species is thought to comprise *Culex Melanoconion* mosquitoes, specifically *Cx. (Mel.) portesi*, from which CATUV has been consistently isolated within Pará State since its original discovery [43]. Additional arthropods associated with CATUV isolations within the Brazilian Amazon include *Cx. declarator*, *Cq. venezuelensis*, *An. nimbus*, and one record of *Ixodes* ticks [35,41,341]. Rodents and/or marsupials are thought to be the primary vertebrate hosts for CATUV [43,292]. Additional isolations of Catu virus have been reported from sentinel capuchin monkeys, sentinel rodents, opossums, and bats within the Brazilian Amazon [35,41,43,292]. A recent serological study found CATUV antibodies within both wild opossums and rodents in Pará State, indicating that this virus continues to circulate within these sylvatic hosts [72]. Disease symptoms include mild fever, headache, body pain and weakness, dizziness, arthralgia, photophobia, and malaise, with recovery occurring within 5 days [43,71]. In Brazil, CATUV has been frequently isolated and antibodies to this virus (typically between 1 and 2% prevalence) have been recorded from multiple human cases within Pará State since its original isolation—including from one human case of a concomitant infection with *Plasmodium vivax* malaria in the 1980s [41,43,71,299,334,342]. Specifically, within a municipality called Breves within Pará State in the Brazilian Amazon, a singular report from the 1990s indicated that 50% of tested individuals have antibodies to both CATUV and a closely related virus, Guama virus (detailed below) [43].

#### 2.3.16. Guamá Virus (GMAV)

Guamá virus (GMAV) was originally isolated from a sentinel capuchin monkey in the Oriboca Forest, Pará State in 1955 [71]. This virus has also been isolated from *Culex Melanoconion* mosquitoes and opossums from French Guiana [338], and GMAV virus was isolated from *Culex Melanoconion* mosquitoes, along with antibodies documented in wild rodents in Suriname [340]. Similar to CATUV, GMAV is thought to circulate naturally within forested environments and transmitted by *Culex Melanoconion* spp. mosquitoes [71]. Specific species that GMAV has been isolated from include *Cx. (Mel.) portesi*, *Cx. (Mel.) spissipes*, *Cx. (Mel.) pedroi*, *Cx. (Mel.) taeniopus*, *Ae. serratus*, *Ae. sexlineatus*, *Sa. chloropterus*, *Ps. albipes*, *Cq. venezuelensis*, *Ms. titillans*, *Li. durhamii*, *Trichoprosopon* spp. mosquitoes along with *Lutzomyia flaviscutellata* sand flies, and *Ixodes* spp. ticks [35,41,43,292,335,341]. Guamá virus is thought to endemically circulate in the Brazilian Amazon and has been found associated with multiple vertebrate animal species. Despite a lack of recent literature on the transmission cycle, GMAV is thought to be maintained within rodents and marsupials [43,292]. In studies, within the Brazilian Amazon (particularly in Pará State), additional isolations were made from sentinel rodents, wild rodents, a wild porcupine, wild birds, and bats, and additional isolations (within the same month of its original isolation) were made from multiple sentinel monkeys [35,41,43,71,292]. Antibodies to GMAV have been documented in sentinel rodents within Pará State since its original isolation as well [41]. Symptoms of human disease resemble those of CATUV: headache, mild fever, body pains, dizziness, photophobia, arthralgia, and sometimes nausea, with a full recovery occurring within 5 days [43,71]. In Brazil, GMAV has also been isolated multiple times from laborers within the Oriboca Forest in the 1950s in Pará State, along with evidence of antibodies to the virus [41,43,71]. It is estimated that human antibody prevalence to GMAV within the Brazilian Amazon is consistently 1–2%; however, residents of a specific municipality in Pará State, Breves, have demonstrated up to 50% prevalence of antibodies to both CATUV and GMAV from the 1990s [43].

#### 2.3.17. Oropouche Virus (OROV)

Oropouche virus (OROV) is notably the most important and most widely distributed orthobunyavirus associated with arthropod vectors and humans in Brazil. This virus was originally isolated from a febrile man in Trinidad in 1955 [343]. Five years following its original isolation in 1960, OROV was isolated both from *Cq. venezuelensis* mosquitoes in Trinidad and from a sloth and pool of *Ae. serratus* mosquitoes in Pará State, Brazil [41,344,345]. The following year, the first recorded epidemic of OROV occurred within the capital city of Pará State, where an estimated 11,000 individuals were affected [346]. Between the time of its original isolation and 1980, outbreaks were restricted to Pará State, Brazil primarily within major cities in a cyclical pattern [43,345]. From 1981 to 1996, outbreaks of OROV began to appear in additional Brazil States covering a larger geographic area including Amazonas, Amapá, Acre, Rodônia, Maranhão, Goiás, and Tocantins; OROV was also isolated from wild monkeys in Minas Gerais State on multiple occasions during this time [347,348]. In addition to the geographic expansion, case numbers within outbreaks increased as well; a singular outbreak in the 1990s impacted over 90,000 patients in 45 days in Rodônia State [297]. It was estimated that over 500,000 people were infected by OROV within the Brazilian Amazon basin between 1961 and 1994 [297]. Oropouche virus began to appear in additional countries as well, with localized outbreaks reported in both Peru and Panama around the same time [345]. Following re-emerging outbreaks of OROV within cities in the Brazilian Amazon, the virus began to expand its geographical range relatively quickly almost yearly [349]: OROV was responsible for multiple outbreaks outside of the country in Haiti in 2014, both Ecuador and Peru in 2016, French Guiana in 2020, and Colombia from 2019 to 2021. As of 2023, OROV has been documented within various vertebrate animals and mosquitoes in most of South America and some Central American countries [349]. Currently, at the time of this manuscript being written (2024–2025), the largest OROV outbreak is occurring in South and Central America. This outbreak originated in the Brazilian Amazon, and sustained local transmission has spread to Peru, Bolivia, Colombia, Cuba, and the Dominican Republic; cases continue to rise in each country, surpassing previous outbreak records for the virus [350]. This year, 2024, is the first year in which Europe and the USA have reported imported OROV cases due to the impact of this outbreak [350,351,352].

Oropouche virus has two transmission cycles: an urban cycle and a sylvatic transmission cycle. Within the sylvatic cycle, the exact arthropod vectors are not entirely clear; however, OROV has been isolated from both *Cq. venezuelensis* (in Trinidad) and *Ae. serratus* mosquitoes in Brazil in forested, sylvatic environments [35,41,43,344,345,348]. Sylvatic vertebrate animal hosts may include sloths, monkeys, rodents, and birds as antibodies to OROV isolations have been reported in all of these animals; however, the mechanism for transmission among these animals is still unknown [35,284,345,353,354,355,356,357]. Sporadic viral isolations have been reported within Brazil from sloths and monkeys [35,346,358]. Antibodies to OROV have also been recently reported in both sheep (Mato Grosso do Sul and Mato Grosso States) and water buffaloes (Pará State); however, their roles in the transmission cycle are unknown [148,303]. Within the urban cycle, more anthropophilic arthropods commonly found in urban epidemics in Brazil like the biting midge *Culicoides paraensis* and *Cx. quiquefasciatus* are implicated as vectors [35,43,345,359]. *Culicoides paraensis* biting midges are thought to be the primary vectors instead of mosquitoes as the biting midges are commonly found infected during epidemics [292,349]. Humans are considered the links between the sylvatic and urban cycles as humans will enter the sylvatic habitat, become infected, and subsequently move into an urban area with high enough viremia to continue transmission as the primary vertebrate hosts of the urban cycle [345,349]. It is still unknown whether humans act as amplification hosts during the urban cycle or whether human movement is the lynchpin to large outbreaks [349,360,361]. Of special concern is São Paulo State as multiple cases of imported OROV-infected patients have traveled into this region and sustained local transmission is possible as the epidemic/urban vectors are present throughout the country [292].

Oropouche virus disease presents clinically as a generally mild and self-limiting disease with symptoms including the abrupt onset of fever, chills, headache, arthralgia, myalgia, vomiting and nausea, and photophobia [43,345,349,350,362,363,364]. Less common symptoms may also occur including epigastric pain, diarrhea, rash, anorexia, retro-orbital pain, conjunctival congestion, and hemorrhagic complications [43,349,362,363]. Central nervous system involvement may also occur, although rarely, in patients, resulting in severe headache, neck stiffness, meningitis, or meningoencephalitis [43,345,349,350,365,366,367]. Disease typically presents with an acute phase lasting 2–4 days, followed by a short remission and return of symptoms with less intensity 7–10 days following initial onset [345,350,364,366]. Long-term sequalae are not common but have been reported in some recent cases, with patients reporting persistent myalgia and general body weakness for up to 1 month [364]. More research is needed to understand this novel transmission route for OROV, but this complicates public health efforts in battling this arbovirus.

Recent circulation: The first fatalities from OROV disease were reported from Brazil in 2024: two women, both from Bahia State [368,369]. A 2024 OROV outbreak led to the first descriptions of OROV infection in pregnant women and spontaneous abortion and microcephaly along with evidence of vertical transmission of the virus from mother to baby in Brazil [370,371].

### 2.4. Phenuviridae

The Phenuviridae family of viruses includes a vast diversity of pathogens capable of causing disease in humans, livestock, invertebrates, and agricultural crops [372,373]. The genera *Bandavirus* and *Phlebovirus* are associated with arthropod vectors such as mosquitoes, sand flies, and ticks, and infect mammals and humans, causing clinical disease [373]. Phenuviruses are enveloped, negative-sense RNA viruses. Only one phenuivirus is associated with arthropod vectors and human infection and found within Brazil, Itaporanga virus. A general depiction of the *Phenuviridae* transmission cycles and vertebrate animal associations is shown in Figure 2.

#### Itaporanga Virus (ITPV)

Itaporanga virus (ITPV) was originally isolated from a sentinel mouse in Itaporanga, São Paulo, Brazil in 1962 [374]. Very little is known about ITPV’s ecology and natural transmission cycle; however, it has been isolated from several mosquito species including *Cx. Melanoconion caudelli* and *Cq. venezuelensis* in the Brazilian Amazon, specifically Pará State [30,35,41]. Itaporanga virus has also been isolated from *Cx. albinensis* mosquitoes in neighboring French Guiana [375]. Vertebrate animals from which ITPV has been isolated include opossums, sentinel rodents, sentinel and wild birds, and sentinel monkeys [30,35,41]. Antibodies to ITPV have also been recorded in opossums within Pará State in the 1950s–1960s [41]. The primary arthropod vector and primary vertebrate animal hosts have not been identified. No record of human disease has been reported to be caused by ITPV; however, a relatively large serological survey within Pará State Brazil in the late 1980s found that 0.8% of persons tested had antibodies to ITPV [45].

Recent circulation: There has been no recent literature on IPTV circulation within Brazil through either serological or molecular detection in humans, animals, or mosquitoes. The last original data were collected in the 1980s.

### 2.5. Rhabdoviridae

Rhabdovirudae includes multiple genera of viruses that can infect plants and animals (many livestock and agriculturally important species), mostly all transmitted through arthropod vectors including ticks, mosquitoes, and sand flies [376]. The arguably most studied rhabdovirus is rabies virus, a non-arthropod-borne virus. Rhabdoviruses are enveloped and non-enveloped negative-sense RNA viruses. A general depiction of the *Rhabdoviridae* transmission cycles and vertebrate animal associations is shown in Figure 2.

#### Jurona Virus (JURV)

Jurona virus (JURV) genetically lies within the vesicular stomatitis serogroup and is the only known arbovirus capable of causing human disease and associated with mosquitoes. This virus was originally isolated from a febrile man in Costa Marques, Rodônia State, Brazil in 1962 [43,297]. Jurona virus was later isolated from a pool of *Hg. janthinomys* mosquitoes within Pará State along the Belem–Brasilia highway between the 1950s and the 1960s [30,35,41,43]. There have been no records of this virus within any other vertebrate animals, and no serological surveys have found antibodies within animals or humans within or outside of Brazil. A highly closely related rhabdovirus was isolated from birds within the northeastern United States of America in the early 2000s, possibly suggesting an avian host [377]. The singular case of human disease was described as febrile, blood smears were negative for malaria, and no other symptoms were recorded at that time [43,297].

Recent circulation: There has been no recent literature on JURV circulation within Brazil through either serological or molecular detection in humans, animals, or mosquitoes.

**Table 1 microorganisms-13-00650-t001:** Arboviruses reported in Brazil that are associated with mosquitoes and humans.

										Original Collection Source Information			References
Family	Genus	Antigenic Group	Name	Abbreviation	Natural Host or Animal Associations ^†^	In Amazon Forest	Associated Arthropods ^‡^	Human Disease	Virus Isolated or Serology	Date	Host Source	Location	
**Togaviridae**													
	Alphavirus	Group A	Mayaro	MAYV	human; **monkeys**; wild rodents; opossums; sloths; equines; lizards; wild birds	Y	***Haemagogus* spp.** including ***Hg. janthinomys***; *Ae. aegypti*; *Sabethini* spp.; *Culex* spp. including *Cx. quinquefasciatus*; *Gigantolaelaps* spp.; *Ixodes* spp.	febrile	positive isolation and serology	1954	human male	Mayaro County, Trinidad	[34,35,36,41,42,43,50,52]
	Alphavirus	Group A	Chikungunya	CHIKV	**human; monkeys**	Y	***Ae. aegypti***, ***Ae. albopictus***, *Ae. fluviatilis; Psorophora albiguenu*, *Ps. ferox; Cx. quinquefasciatus*; *Wyeomyia bourrouli*	febrile	positive isolation and serology	1953	human female	Liteho, Newala District, Tanzania	[49,50,51,52,53,56]
	Alphavirus	Group A	Eastern Equine encephalitis	EEEV (South American strains)	human; equines; **wild birds**; **wild rodents**; opossums; wild boars	Y	***Culex* spp.** including ***Cx. (Mel.) taeniopus***, ***Cx. (Mel.) pedroi***, ***Cx. (Mel.) spissipes***, *Cx. quinquefasciatus*; *Ae. aegypti*, *Ae. albopictus*, *Ae. taeniorhynchus*	encephalitic	positive isolation and serology	1933	horse	Delaware, Virginia, Maryland, USA	[30,41,43,45,51,52,63,64,66]
	Alphavirus	Group A	Western Equine encephalitis	WEEV	human; equines; **wild birds**; sentinel rodents; opossums	Y	***Culex* spp.** including *Cx. taeniopus*, *Cx. portesi*, *Cx. pedroi; Ae. fulvus*	encephalitic	positive serology	1930	horse	Merced County, CA, USA	[30,35,41,43,45,76,79,80,81]
	Alphavirus	Group A	Mosso das Pedras	VEEV IF	human; **wild rodents;** bats	Y	***Culex* spp.**	febrile	positive isolation and serology	1978	*Cx.* mosquitoes and wild bat	Sitio de Mosso das Pedras, São Paulo, Brazil	[99]
	Alphavirus	Group A	Mucambo	MUCV (VEEV IIIA)	human; sentinel capuchin monkey; sentinel rodents; **wild rodents**; opossums; wild birds	Y	*Culex* spp. including ***Cx. (Mel.) portesi***; *Ae. hortatory*, *Ae. serratus; Haemagogus* spp.; *Mansonia* spp.; *Cq. venezuelnsis*; *Sabethes* spp.; *Uranotaenia geometrica*	febrile	positive isolation and serology	1954	sentinel capuchin monkey	Oriboca Forest, Amazonas, Brazil	[41,43,101,102,103,104]
	Alphavirus	Group A	Pixuna	PIXV (VEEV IV)	opossums; **wild rodents**; wild birds	Y	*An. nimbus; Trichoprosopon digitatum*	febrile	positive isolation and serology	1985	*Tr. digitatum* mosquitoes	Belem, Para, Brazil	[30,43,106]
	Alphavirus	Group A	Una	UNAV	human; wild rodents, horses, cows; birds	Y	*Ps. ferox*, *Ps. albipes*, *Ps. lutzii*; *Ae. serratus*, *Ae. fulvus*, *Ae. leucocelaenus*; *An. nimbus*; *Coquillettidia arribalzaga*; *Culex* spp.; *Wyeomyia* spp.	febrile	positive serology	1959	*Ps. ferox* mosquitoes	Belem, Para, Brazil	[30,35,41,109]
	Alphavirus	Group A	Aura	AURAV	human; marsupials; wild rodents; horses	Y	*Culex* spp.; *Ae. serratus*	unknown	positive serology	1959	*Cx. Melanoconion* mosquitoes	Belem, Pará, Brazil	[30,35,41,110,117]
**Flaviviridae**													
	Flavivirus	Group B	Dengue 1	DENV-1	**human; monkeys**; sloths; equines	Y	***Aedes aegypti***, *Ae. albopictus*; *Haemagogus leucocelaenus*	febrile	positive isolation and serology	1943	human male	Nagasaki, Japan	[30,119,120,129,132,378]
	Flavivirus	Group B	Dengue 2	DENV-2	**human; monkeys**; sloths; equines	Y	***Aedes aegypti***, *Ae. albopictus; Culex* spp. including *Cx. vaxus*	febrile	positive isolation and serology	1944	human male	New Guinea	[30,129,132,379,380]
	Flavivirus	Group B	Dengue 3	DENV-3	**human; monkeys**; sloths; equines	Y	***Aedes aegypti***, *Ae. albopictus*	febrile	positive isolation and serology	1954	human female and *Aedes aegypti* mosquitoes	Manila, Philippines	[30,124,129,132]
	Flavivirus	Group B	Dengue 4	DENV-4	**human; monkeys**; sloths; equines	Y	***Aedes aegypti****; Culex quinquefasciatus*, *Cx. bidens*, *Cx. interfor*; *Psorophora varipes*, *Ps. albigenu; Sabethes chloropterus*	febrile	positive isolation and serology	1954	human female and *Aedes aegypti* mosquitoes	Quezon City, Manila, Philippines	[30,124,129,132,135]
	Flavivirus	Group B	Yellow Fever	YFV	**human; monkeys**; water buffaloes	Y	***Aedes aegypti***, *Ae. albopictus*, *Ae. scapularis*, *Ae. taeniorhynchus*, *Ae. serratus*; ***Haemagogus janithinomys***, ***Hg. leucocelaemus***, *Hg. albomaculatus*, *Hg. capricornii*, *Hg. spegazzinii*; ***Sabethes chloropterus***, *Sa. soperi*, *Sa. cyaneus*, *Sa. glaucodaemon*, *Sa. albiprivus*, *Psorophora ferox*	febrile	positive isolation and serology	1927	human male	Kpeve Village, Ghana	[30,43,144,147,148,172]
	Flavivirus	Group B	Zika	ZIKV	**human; monkeys**	Y	** *Aedes aegypti; Ae. albopictus* ** *; Haemagogus leucoelaenus; Culex quinquefasciatus; Anopheles cruzii; Limatus durhamii; Wyeomyia confusa*	febrile with neurological and pregnancy complications	positive isolation and serology	1947	sentinel rhesus monkey	Zika Forest, Entebbe, Uganda	[51,53,153,154,156,157,172,379,381,382]
	Flavivirus	Group B	Ilhéus	ILHV	human; sentinel monkeys; bats; rodents; marsupials; sloths; edentate mammals; water buffaloes; cattle; equines; pigs; reptiles; **wild birds**	Y	*Psorophora* spp. including ***Ps. ferox***, *Ps. albipes*, *Ps. lutzii*; *Aedes* spp. including *Ae. aegypti*, ***Ae. serratus***, *Ae. fulvus*, *Ae. leucocelaenus*, *Ae. scapularis*; *Culex* spp. including *Cx. portesi*, *Cx. coronator*; *Sabethes chloropterus*.; *Haemagogus* spp. including *Hg. leucocelaemus*; *Trichoprosopon* spp.	febrile and encephalitic	positive isolation and serology	1944	*Aedes* and *Psorophora* mosquitoes, including *Ae. serratus* and *Ps. ferox*	Ilhéus, Brazil	[30,41,43,52,72,101,148,175,329,383]
	Flavivirus	Group B	West Nile	WNV	human; equines; **wild birds**; domestic birds	Y	***Culex* spp. including *Cx. quinquefasciatus* and *Cx. pipiens***	encephalitic	positive isolation and serology	1937	human female	Omogo, West Nile District, Uganda	[195,213,228,232,237,384]
	Flavivirus	Group B	St. Louis Encephalitis	SLEV	human; equines; **wild rodents**; **wild birds**; sentinel rodents, sentinel chickens; sentinel monkeys; wild monkeys; opossums; sloths; water buffaloes	Y	*Sabethes* spp. including *Sa. belisarioi*; *Culex* spp. including ***Cx. coronator*, *Cx. declarator*, *Cx. nigripalpus*, *Cx. pipiens*, *Cx. quinquefasciatus***; *Aedes* spp.; *Mansonia* spp.; *Gigantolaelops* spp.	Febrile and encephalitic	positive isolation and serology	1933	human	St. Louis County, MI, USA	[30,41,43,119,148,233,245,253,254,255,261,385]
	Flavivirus	Group B	Bussuquara	BSQV	human; sentinel howler monkeys; sentinel rodents; **wild rodents**; water buffaloes	Y	*Culex* spp. including *Cx. declarator*, *Cx. portesi*, *Cx. pedroi*, *Cx. taeniopus*; *Cq. venezuelensis*, *Mansonia titillans*	febrile	positive isolation	1956	sentinel howler monkey	Belem, Pará, Brazil	[30,35,41,42,101,148,262]
	Flavivirus	Group B	Rocio	ROCV	human; **wild birds;** sentinel rodents; wild rodents; equines; water buffaloes; bats; marsupials	Y	*Psorophora* spp. including ***Ps. ferox***; *Aedes* spp. including *Ae. scapularis*; *Cx. portesi*, *Cx. quinquefasciatus*	encephalitic	positive isolation and serology	1975	human male	Iguape County, Sao Paulo, Brazil	[101,106,233,267,269,270,271]
	Flavivirus	Group B	Cacipacore	CPCV	human; monkeys; birds; wild rodents; equines; water buffaloes	Y	*Aedes aegypti; Anopheles* spp.; *Culex* spp.; *Amblyomma cajennense*	unknown	positive isolation and serology	1977	black-faced ant bird	Oriximina, Para, Brazil	[30,192,279,280,285]
**Peribunyaviridae**													
	Orthobunyavirus	Anopheles A	Tacaiuma	TCMV	human; sentinel **capuchin monkeys**; equines; small ruminants; wild rodents; birds; bats; water buffaloes	Y	*Aedes triannulatus*, *Ae. scapularis*; ***Anopheles*** **spp. including** ***An. cruzii***; *Haemagogus* spp. including *Hg. janthinomys*	febrile	positive isolation and serology	1955	sentinel capuchin monkey	Oriboca Forest, Para, Brazil	[35,43,71,292,294]
	Orthobunyavirus	Anopheles B	Boraceia	BORV	Human; cattle; equines; dogs; domestic and wild birds; wild rodents; marsupials	N	***Anopheles* spp. including *An. cruzii*** *; Wyeomyia pilicauda*	unknown	positive isolation and serology	1962	*An. cruzii* mosquitoes	Casa Grande, São Paulo, Brazil	[292,300,301]
	Orthobunyavirus	Bunyamwera	Tucunduba	TUCV	human	Y	*Anopheles* spp. including *An. nimbus*; *Ae. fulvus*, *Ae. scapularis*, *Ae. argyrothorax*, *Ae. serratus*, *Ae. sexlineatus*, *Ae. seplemstriatus*; *Psorophora ferox*; *Hg. leucoceiaenus*; *Cx. coronator*, *Cx. ocellatus*; *Limatus flavisetosus*, *Li. durhanii*; ***Wyeomyia*** **spp. including *Wy. aporonomo***; *Sabethes* spp. including *Sa. quassicyaneus*, *Sa. interinedius*; *Trichoprosopon* spp. including *Tr. digitatum*	encephalitic and febrile	positive isolation and serology	1955	*Wy.* spp. mosquitoes	Oriboca Forest, Para, Brazil	[35,41,43,292,297]
	Orthobunyavirus	Bunyamwera	Maguari	MAGV	human; sentinel rodents; wild rodents; equines; sheep; water buffaloes	Y	*Ae. sexlineatus*, *Ae. fulvus*, *Ae. leucocelaenus*, *Ae. scapularis*, *Ae. serratus*; *Cx. pedroi*; *An. nimbus*; *Psorophora* spp. including *Ps. ferox*, *Ps. albipes*; *Mansonia* spp., *Limatus* spp.; *Wyeomyia* spp.	febrile	positive isolation and serology	1957	mixed mosquito pool	Utinga Forest, Belem, Para, Brazil	[41,71,292,294,295]
	Orthobunyavirus	Bunyamwera	Anhembi	AMBV	human; wild rodents; wild birds	N	*Wyeomyia pilicauda; Trichoprosopon pallidiventer*	unknown	positive serology	1965	*Wy. pilicauda* mosquitoes	Casa Grande, Brazil	[292,310]
	Orthobunyavirus	Bunyamwera	Macauã	MCAV	human; wild rodents; wild birds	Y	*Sabethes soperi*	unknown	positive serology	1976	*Sa. soperi* mosquito	Sena Madureira, Acre, Brazil	[35,42,106,292]
	Orthobunyavirus	California	Guaroa	GROV	human; **wild birds**; water buffaloes; sheep	Y	***Anopheles* spp.** including *An. triannulatus*, *An. nunestovari*	febrile	positive isolation and serology	1956	human female	Guaroa, Colombia	[43,292,303,313]
	Orthobunyavirus	California	Serra do Navio	SDNV	human; wild rodents; opossums	Y	*Aedes fulvus*	unknown	positive serology	1966	*Ae. fulvus* mosquito	Belem, Para, Brazil	[42,79,292,318]
	Orthobunyavirus	Group C	Apeu	APEUV	human; sentinel capuchin monkeys; opossums; wild non-human primates; sentinel rodents; sheep	Y	***Culex* spp.** including *Cx. aikenii*, *Cx. portesii; Aedes arborealis*, *Ae. septemstriatus*	febrile	positive isolation and serology	1955	sentinel capuchin monkey	Oriboca Forest, Para, Brazil	[30,41,43,71,292,303,320,321]
	Orthobunyavirus	Group C	Caraparu	CARV	human; sentinel capuchin monkeys; **wild rodents**; sentinel rodents; bats; equines; water buffaloes; wild birds	Y	***Culex* spp.** including *Cx. aikenii*, *Cx. vomerifer*, *Cx. portesi*, *Cx. sacchettae*, *Cx. spissipes*, *Cx. coronator*, *Cx. nigripalpus*, *Cx. accelerans*, *Cx. amazonensis; Ae. scapularis*, *Ae. serratus*; *Wyeomyia medioalbipes*; *Sabethes* spp.; *Limatus durhamii*; *Ps. ferox*	febrile	positive isolation and serology	1956	sentinel capuchin monkey	Belem, Para, Brazil	[35,41,43,71,292,319,326]
	Orthobunyavirus	Group C	Itaqui	ITQV	human; sentinel capuchin monkeys; sentinel rodents; **wild rodents**; opossums	Y	***Culex* spp.** including *Cx. vomerifer*, *Cx. portesi*, *Cx. aikenii*, *Cx. spissipes*	febrile	positive isolation and serology	1956	sentinel capuchin monkey	Belem, Para, Brazil	[35,41,43,292,330]
	Orthobunyavirus	Group C	Marituba	MTBV	human; sentinel capuchin monkeys; sentinel rodents; opossums	Y	***Culex* spp.** including *Cx. aikenii*, *Cx. portesi*	febrile	positive isolation and serology	1954	sentinel capuchin monkey	Oriboca Forest, Para, Brazil	[35,43,71,292]
	Orthobunyavirus	Group C	Murucutu	MURV	human; sentinel capuchin monkeys; sentinel rodents; wild rodents; opossums; sloths; birds	Y	***Culex* spp.** including *Cx. (Mel.) caudelli*, *Cx. aikenii. Cx. portesi*, *Cx. vomerifer*; *Sabethini* spp.; Ixodid ticks	febrile	positive isolation and serology	1955	sentinel capuchin monkey	Oriboca Forest, Para, Brazil	[41,43,71,106,292]
	Orthobunyavirus	Group C	Oriboca	ORIV	human; sentinel capuchin monkeys; opossums; sentinel rodents; **wild rodents**; sheep	Y	***Culex* spp.** including *Cx. (Mel.) portesi*, *Cx. (Mel.) spissipes*, *Cx. (Mel.) caudelli; Ps. ferox; Sabethini* spp.; *Aedes* spp. including *Ae. arborealis*, *Ae. serratus*, *Ae. argyrothorax*; *Cq. arribalgagae*, *Cq*, *venezuelnsis*, *Mansonia* spp.; *Wyeomyia* spp.	febrile	positive isolation and serology	1954	sentinel capuchin monkey	Oriboca Forest, Para, Brazil	[35,41,43,71,292,303]
	Orthobunyavirus	Guamá	Catu	CATUV	human; sentinel capuchin monkey; sentinel rodents; **wild rodents**; opossums; bats	Y	***Culex* spp.** including *Cx. (Mel.) portesi*, *Cx. declarator*; *An. nimbus; Cq*, *venezuelensis; Ixodes* spp.	febrile	positive isolation and serology	1955	human male	Oriboca Forest, Para, Brazil	[35,43,71,292]
	Orthobunyavirus	Guamá	Guamá	GMAV	human; sentinel capuchin and howler monkeys; sentinel rodents; **wild rodents**; wild birds; opossums; bats; porcupines	Y	***Culex* spp.** including *Cx. (Mel.) portesi; Sabethes chloropterus; Cx. (Mel.) spissipes*, *Cx. (Mel.) pedroi*, *Cx. (Mel.) taeniopus; Aedes* spp. including *Ae. serratus*, *Ae. sexlineatus*; *Sabethes* spp. including *Sa. chloropterus*; *Ma. titillans*.; *Li. durhamii*; *Psorophora* spp. including *Ps. albipes*; *Cq*, *venezuelensis*; *Trichoprosopon* spp.; *Lutzomyia flaviscutellata*; *Ixodes* spp.	febrile	positive isolation and serology	1955	sentinel capuchin monkey	Oriboca Forest, Para, Brazil	[35,41,43,71,292]
	Orthobunyavirus	Simbu	Oropouche	OROV	**human**; **sloths**; domestic and **wild birds**; sentinel birds; **wild rodents; monkeys**; sheep; water buffaloes	Y	***Ae. serratus****; Culex spp. including **Cx.*** ***quinquefasciatus;*** ***Cq. venezuelensis; Culicoides spp.** including **Cu. paranensis***	encephalitic and febrile	positive isolation and serology	1955	human male	Sangre Grande, Trinidad	[41,43,52,148,284,292,303,343,345,346,353,354,355,356,357,358,386]
**Phenuiviridae**													
	Phlebovirus	Phlebotomus Fever	Itaporanga	ITPV	human; sentinel rodents; opossums; sentinel and wild birds; sentinel monkeys	Y	*Culex Melanoconion* spp. including *Cx. caudelli*; *Coquillettidia venezuelensis*	unknown	positive serology	1962	sentinel Swiss mouse	Itapiranga, São Paulo, Brazil	[30,35,41,45,387]
**Rhabdoviridae**													
	Vesiculovirus	Vesicular Stomatitis	Jurona	JURV	human	Y	*Haemagogus janthinomys*	febrile	positive isolation and serology	1962	Human male	Costa Marques, Rodônia State, Brazil	[30,35,41,43,297]

**^†^** Bold animal associations are reservoir and/or amplification hosts if known. **^‡^** Bold arthropod associations are vectors if known.

## 3. How Has Arbovirus Emergence and Re-Emergence Correlated with Anthropogenic Landscape Changes?

Evidence for vector-borne disease emergence related to anthropogenic factors has increased over time, particularly due to the impacts of deforestation, urbanization, and agricultural expansion. These activities lead to habitat degradation and local biodiversity changes, which in turn has increased vector-borne disease cases and influenced the emergence of new diseases [388]. In regard to deforestation—one of the most influential activities, particularly in the Brazilian Amazon—this involves a transformation of natural ecosystems, which leads to the alteration of arthropod vector species’ natural habitats. Arthropod vectors will subsequently travel to find favorable conditions, sometimes closer to urban areas, increasing the likelihood for human–vector interactions [388]. This habitat alteration has proved especially crucial for the eco-epidemiology of viruses like the dengue, West Nile, and Chikungunya viruses [388,389].

Urbanization impacts will continue to influence these interactions as global urban centers increase in size over time. A 68.7% (6.7 billion) population increase is projected by the year 2050 [390]. Urbanization will thus lead to increased deforestation, agricultural expansion, and continued human migration—all previously shown to increase mosquito-borne arboviral diseases [388,389]. Of special concern are mosquitoes adapted to humans, those that are primarily anthropophilic, like *Ae. aegypti* and *Ae. albopictus*, but additional species within the *Culex* and *Anopheles* genera have also been linked to urbanization and increased risk for arboviral disease [390]. Moreover, anthropogenic land use changes have also been linked with effects on mosquito host-seeking behavior and, consequently, the risk of virus transmission to humans [391,392]. Additionally, deforestation and human migration lead to increased risks for zoonotic disease spillover as ecosystem disturbances lead to host animal availability and proximity differences, changes in vector feeding habits, vector diversity and dispersal, alterations of vegetation and vector resting behaviors, and micro-climate changes even impacting the life histories of arthropod vectors [392,393,394,395,396,397,398]. This has been the case specifically for multiple arboviruses within Central America, i.e., Zika and dengue viruses have been unequivocally linked to human-maintained anthropogenic changes within the natural environment like illegal cattle ranching following deforestation, human migration, road construction, mining, plantation agriculture, and urbanization [25].

In Brazil, there are multiple examples of human-maintained anthropogenic change and subsequent vector-borne disease incidence changes. This has been documented within the country since the early 1970s, following the construction of the Trans-Amazon Highway in the State of Pará, Brazil [342]. Within three years of construction, investigators found that 10% of the migrant workers (n = 308) at that time had evidence of positive serologic conversion to different arboviruses that were known at the time [342]. And since this time, more than just the Trans-Amazon Highway has been built. Multiple reviews and manuscripts provide evidence that human-maintained anthropogenic change is linked to increases in multiple pandemic-level arboviruses including the Chikungunya, dengue, Zika, and West Nile viruses [6,399].

An example of an arbovirus that has historically been impacted by human-maintained anthropogenic change in Brazil is yellow fever virus. Road construction within the Brazilian Amazon showed an increase in YFV activity in humans decades prior, and evidence of continued deforestation and landscape changes is associated with YFV in the country [342]. The more recent outbreaks of YFV within Brazil not only have reached new case levels never before seen in the country, but disease and virus circulation have also been documented in regions of the country where virus circulation had not been seen in over 60 years—primarily along the Atlantic Coast [400,401]. Multiple studies hypothesized that forest fragmentation adjacent to urbanized settings along the Atlantic Coast were associated with this increase in the number of YFV cases [402,403]. Forested fragments within and adjacent to urban landscapes provide structural pathways for sylvatic YFV vertebrate host animals (non-human primates) to move close to humans, along with the presence of bridge vectors capable of feeding on both non-human primates and humans [401,402,403,404]. A recent study found that an 85% increase in the occurrence of YFV events in both humans and non-human primates within São Paulo State was significantly associated with highly fragmented forest cover, with intermediate levels of forest fragmentation also contributing to YFV dispersion and YF cases [401]. In alignment with this analysis, another recent study found that vegetation fragmentation significantly decreases the time needed for YFV spread within this same region [405]. Although the urban transmission cycle has not occurred following these sylvatic outbreaks, the risk is not zero as the cosmopolitan and anthropophilic vector, *Ae. aegypti*, is present in the urban settings adjacent to fragmented forests [401].

A current example of an arbovirus’s public health impacts associated with human-mediated anthropogenic change with Brazil is oropouche virus and the ongoing outbreak. The ongoing outbreak is both the largest recorded OROV outbreak in the 21st century and the most geographically spread OROV epidemic ever documented [406]. A novel viral reassortment lineage has been identified in this current outbreak, and one study investigated this novel reassortment’s geographic dissemination within the Brazilian Amazon, developing a live view of viral movement within an epidemic [406]. A significant proportion of OROV migrations (31%) within the region was attributed directly to human migration and movement throughout the region wherein infected individuals moved from rural areas (where they were initially infected) into urban areas without OROV circulation, thus introducing the virus to an urban transmission cycle [406]. A second study found specific agricultural activities such as banana and cocoa cultivation, along with reductions in forest cover, to be significantly associated with OROV range expansion, utilizing the ecological niche modeling of the current epidemic in the Amazon [407]. Furthermore, most municipalities with previous OROV outbreaks (from 2022 to 2023) were all adjacent to new frontiers of deforestation and expansions of agricultural development within the Amazon from 2017 to 2021 [401,408,409]. Continued surveillance and efforts to mitigate the outbreak are crucial in rural areas, where urbanization and deforestation are planned as OROV outbreaks will most likely continue to occur [406,407,410,411].

As human-maintained anthropogenic change occurs throughout Brazil, continued outbreaks of known and unknown arboviruses through the emergence or resurgence of disease are a matter of when, not if. There are a few arboviral candidates with important public health impacts that may cause future epidemics due to the influence of human-maintained anthropogenic change, including the Mayaro and Rocio viruses, based on their past outbreak epidemiology and ecologies. Both the Mayaro and Rocio viruses can cause significant disease in humans, and although both have not been associated with significant outbreaks with large geographical spreads, both viruses are thought to be significantly underreported, and there are multiple associated mosquito vectors that feed on a variety of animals, which are sensitive to landscape change [46,133,277]. The sequence of events that has led to the resurgence and emergence of both YFV and OROV seems to align well with both MAYV and ROCV, which are known to circulate within sylvatic vertebrate animals and mosquitoes; thus, MAYV and ROCV may already be increasing in circulation at an unknown level today [412].

Public health action must be taken in order to help circumvent current and future arbovirus outbreaks in Brazil. Programs focused on the control and prevention of arboviral disease must not rely on a singular tool (i.e., solely education or chemical insecticides), but rather, on an integrated strategy utilizing traditional mosquito surveillance, host animal surveillance, chemical insecticides, environmental management, public education, reliable and accurate diagnostics and communication among public health officials, and genomic surveillance, and even exploring novel genetic or biological control methods is necessary [413,414]. The use of traditional methods alone has failed in the past to prevent and control mosquito-borne disease in this country [414]. Although a detailed description of these action steps is beyond the scope of this review, it is important to note some recent successful arbovirus control or prevention campaigns for arboviruses. A simple yet effective strategy is strengthening communication among epidemiologists and health officials during surveillance to better prepare for outbreaks. One study evaluated a surveillance system within Paraná State, where diagnoses of dengue virus were made significantly more efficient and accurate through a formal communication-based surveillance system among public health managers and officials [415]. Evidence-based decisions were made on a faster, more efficient timeline and resources were more successfully allocated to higher-risk areas compared to the situation before this system was in place [415]. The use and release of *Wolbachia*-infected *Ae. aegypti* mosquitoes (a biological control method) as an intervention for dengue virus has had remarkable success in Oceania—notably in Australia and Indonesia [416,417]. Recently, this strategy was successful in reducing dengue incidence in two locations of Rio de Janeiro, Brazil by an estimated 38% and 69% [418,419]. In addition, public education is one of the most important tools in the proverbial mosquito-borne arbovirus toolbox. One study evaluated community knowledge on Zika virus following the ZIKV pandemic in 2015–2016 from two epidemiologically distinct municipalities to understand how country-wide efforts during response impacted public behaviors, i.e., knowledge, attitudes, and practices. This study found that overall awareness of sexual transmission risk for ZIKV was low, targeted educational campaigns for women did not improve personal protective measures, and household barriers to control mosquitoes included a lack of buy-in from community and municipal levels [420]. Future public health education campaigns should learn from this pandemic to garner additional support on the local level for encouraging personal protection against mosquito bites and increase educational campaigns for men [420]. Lastly, vaccine development and vaccine implementation strategies are key for mosquito-borne arbovirus prevention. Although there are only a few vaccines available for human arboviruses, there are many currently in development [413]. In short, vaccines work and have been credited in preventing massive arbovirus outbreaks, specifically from yellow fever in Brazil [421]. Public health funding on an international level should continue to advance vaccine development for the most impactful arboviruses first, followed by additional viruses known to cause human disease.

## 4. Conclusions

Continued human-maintained landscape changes throughout Brazil will result in the emergence and resurgence of additional vector-borne arboviruses and disease cases. There are hundreds of arboviruses known to circulate within the country, and the scientific community has only scratched the surface in regard to identifying those viruses that are capable of causing disease in humans and are associated with mosquitoes. This review has aimed to compile the existing information on these arboviruses regarding their history, ecological niches, associated vertebrate animal hosts and arthropods, and current geographic spread—with a certain nod to the importance of human-maintained anthropogenic change and its interaction with these viruses and their public heath potential. Surveillance and resources are needed to continue to monitor for the presence of these viruses within areas where significant anthropogenic change is occurring in order to prevent future cases or the next pandemic.

## Figures and Tables

**Figure 1 microorganisms-13-00650-f001:**
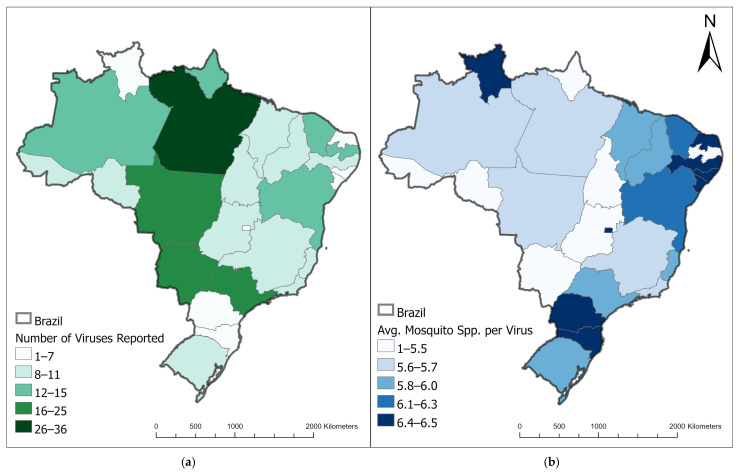
The diversity of viruses associated with humans and mosquitoes reported in Brazil varies across the country. (**a**) displays the diversity of viruses reported within each state (detected through both serological and molecular methods), where the darkest green color indicates a higher virus burden. (**b**) displays the average number of mosquito species associated with each virus found within each state, where the darker blue color indicates a higher average number of different associated mosquito species per virus reported.

**Figure 2 microorganisms-13-00650-f002:**
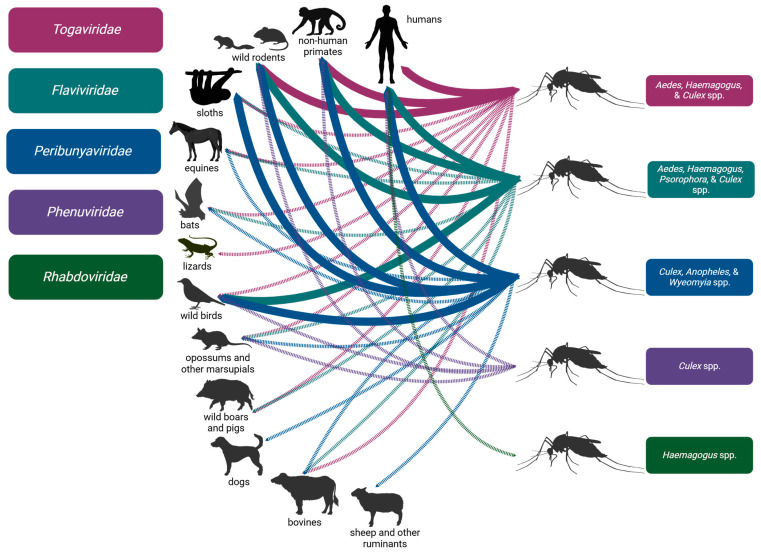
Transmission cycles and associations of various vertebrate animals and most important mosquito vectors across the five arbovirus families described in this manuscript. Sold, thicker lines depict evidence of associated animals as virus hosts within the transmission cycle; thinner dashed lines depict possible animals as hosts, but no evidence of hosts has been reported, only associations (serological evidence of infection). Created in BioRender (2025). https://BioRender.com/u96t994 (accessed on 21 February 2025).

## Data Availability

No new data were required for this review manuscript.

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
