# Peer review of "(Re)Emerging Arboviruses of Public Health Significance in the Brazilian Amazon"

_microorganisms, 2025, doi:10.3390/microorganisms13030650_

Round 1

Reviewer 1 Report

Comments and Suggestions for Authors

The study provides a comprehensive review of arboviruses, focusing on their transmission cycles, associated vectors, and vertebrate hosts, with particular emphasis on the Amazon region in Brazil. The research documents the diversity of over 210 arboviruses identified in Brazil, with 39 linked to humans and mosquitoes. Detailed descriptions of viruses such as Mayaro, Chikungunya, and Zika are included, providing valuable insights into their epidemiology and public health implications. Furthermore, the study emphasizes the significance of non-human hosts, including primates and other animals, in the sylvatic transmission cycles of arboviruses like yellow fever and Zika. Lastly, the authors highlight the emerging threat of arboviruses spreading beyond the Amazon and stress the importance of robust surveillance and preventive measures to mitigate future outbreaks.

Some suggestions:  

1.       First, this manuscript relies heavily on historical data and serological surveys, so a clearer distinction between recent findings and historical observations would help enhance the context and relevance of the data presented.

2. While the paper outlines the challenges posed by arboviruses, it offers limited specific recommendations for public health policies and interventions. Authors could include actionable steps such as vector control measures, vaccination campaigns, and community engagement strategies to significantly enhance teh findings.

3. Lastly, the addition of visual elements such as maps, graphs, or infographics to depict arboviral distributions and transmission cycles would improve reader engagement and comprehension of the findings.

Reviewer 2 Report

Comments and Suggestions for Authors

This manuscript is well-written, deals with a significant subject and should be published.

However, some issues need to be resolved before publication:

1) What are the aim and or the purpose of this review? Parts 1 (Introduction), 3, and 4 (Conclusions) are well written; the conclusion specifies the aim of the review very accurately. However, part 2 is very problematic. It is basically a grocery list of all arboviruses present in the Brazilian Amazon. It is very long, tidies and the authors lose their readers in this part. I think the viruses should be specified but in a much shorter way and with a strong emphasis and focus on emergence and re-emergence in Brazil and the environmental and climate changes behind the outbreaks of each virus.

2) It is crucial that Table 1 will come at the end of Part 2.
